# SAFE: Finding Sparse and Flat Minima to Improve Pruning

**Dongyeop Lee** [1]   **Kwanhee Lee** [1]   **Jinseok Chung** [1]   **Namhoon Lee** [1]

## Abstract

Sparsifying neural networks often suffers from seemingly inevitable performance degradation, and it remains challenging to restore the original performance despite much recent progress. Motivated by recent studies in robust optimization, we aim to tackle this problem by finding subnetworks that are both sparse and flat at the same time. Specifically, we formulate pruning as a sparsity-constrained optimization problem where flatness is encouraged as an objective. We solve it explicitly via an augmented Lagrange dual approach and extend it further by proposing a generalized projection operation, resulting in novel pruning methods called SAFE and its extension, SAFE$^+$. Extensive evaluations on standard image classification and language modeling tasks reveal that SAFE consistently yields sparse networks with improved generalization performance, which compares competitively to well-established baselines. In addition, SAFE demonstrates resilience to noisy data, making it well-suited for real-world conditions.

## 1. Introduction

Over the past decades, the emergence of computers and the accumulation of digital data have made machine learning a viable tool for everyday applications. However, modern machine learning systems have also grown in complexity with the rapid advances in hardware and databases, demanding significant computational and memory resources. This has led to a surge of interest in strategies to reduce these substantial computational costs.

One major approach is sparsification, which aims to find solutions with mostly zero entries. This has been studied for many years in various large-scale applications in machine learning (LeCun et al., 1989; Hassibi & Stork, 1992), signal processing (Blumensath & Davies, 2009), and statistics (Tibshirani, 1996; Beck & Teboulle, 2009). Notably, the recent success of highly overparameterized deep neural networks in achieving human-level computer vision and natural language processing tasks has drastically scaled these models to proportions never seen before, which has spurred considerable research on sparsifying neural networks (Hoefler et al., 2021). Since the pioneering work of Han et al. (2015), many works have proposed to remove redundant parameters by various tactics ranging from those discussed in the survey work of Hoefler et al. (2021) to their applications to large foundation models (Kwon et al., 2022; Frantar & Alistarh, 2023; Sun et al., 2024). However, it is witnessed that excessive pruning usually results in performance decline due to the reduced capacity, potentially impairing deep learning models from handling tasks of high complexity. Is there a way to restore this performance loss?

We attend to recent studies that have demonstrated that the generalization performance of a model is closely linked to the flatness of the solution landscape (Keskar et al., 2017; Jiang et al., 2020a). This insight has led to the development of techniques such as sharpness-aware minimization (SAM) (Foret et al., 2021), which explicitly regularizes the sharpness of the solution during the optimization process. SAM has been shown to deliver exceptional performance across various domains (Chen et al., 2022; Bahri et al., 2022) and has also demonstrated robustness to label noise (Baek et al., 2024). In light of the SAM's success, there has been growing interest in applying these techniques to model pruning. Research by Na et al. (2022) has explored how sharpness-aware training affects model compressibility, while Peste et al. (2022) and Bair et al. (2023) have examined different strategies to alter this process for pruning, with the goal of enhancing performance and robustness. Nevertheless, the exploration of sharpness-aware techniques within the context of sparsification is still at an early stage. We believe there is considerable potential to integrate these approaches more effectively into the sparsification process, which could lead to significant advancements in the development of efficient and robust deep learning models.

In this work, we aim to achieve exactly that, by proposing to frame it as a sharpness-aware sparsity-constrained optimization problem to simultaneously consider both sharpness and sparsity during the training process. To tackle this, we

[1]POSTECH, South Korea. Correspondence to: Dongyeop Lee <dylee23@postech.ac.kr>.

*Proceedings of the 42nd International Conference on Machine Learning*, Vancouver, Canada. PMLR 267, 2025. Copyright 2025 by the author(s).

use an augmented Lagrangian dual-based approach well established in the optimization literature with convergence guarantees, and propose a new optimization-based pruning method called SAFE. We evaluate SAFE across standard benchmark tasks in image classification and large language model post-training pruning, and compare with existing alternatives to demonstrate that (i) it induces flatness on the sparse solutions, (ii) yielding performance improvements in the resulting sparse models, and (iii) its robustness to label noise, common image noise, and adversarial noise, and its (iv) effectiveness compared to similar sharpness-minimization-inspired pruning techniques. These aspects support the effectiveness and generality of our method, and we conclude by discussing current limitations and potential ideas for future work.

## 2. Background

### 2.1. Sparsity

Throughout the years, various techniques to induce sparsity in machine learning systems have been developed to simplify models for efficiency, interpretability, and generalization. This is usually framed as solving the following optimization problem with sparsity constraint:

$$\min_{\|x\|_0 \leq d} f(x), \qquad (1)$$

where $f$ is the objective we wish to minimize, $x$ is the optimization variable, $\|x\|_0$ denotes the $L_0$-norm, which counts the non-zero entries within $x$, and $d$ is the number of parameters we wish to preserve. The goal is to find a solution $x$ with desired sparsity that minimizes $f$. Exactly solving this optimization problem is challenging due to the combinatorial nature of the $L_0$-norm, as it requires an exhaustive search over all possible configurations of zeros within $x$. Consequently, several approaches have been proposed, such as relaxing the $L_0$ norm as in LASSO (Tibshirani, 1996), or employing advanced optimization techniques, including proximal methods like FISTA (Beck & Teboulle, 2009) and iterative hard thresholding (Blumensath & Davies, 2009). Additionally, strategies like optimal brain damage and surgeon (LeCun et al., 1989; Hassibi & Stork, 1992) have been explored to sparsify multi-layer perceptrons through second-order approximations of the objective function.

The recent success of increasingly large deep neural networks has further accelerated this trend, spurring the development of various methods to sparsify neural networks at different stages of training, each offering distinct advantages depending on the scenario (Hoefler et al., 2021). For instance, pruning before training (Lee et al., 2019; Tanaka et al., 2020; Wang et al., 2020) is advantageous for improving computational efficiency during training by enabling sparse training, while post-training pruning (Frantar & Al-

istarh, 2023; Sun et al., 2024; Kwon et al., 2022) is ideal for enhancing inference efficiency in pre-trained models, particularly when the training process can be too costly due to large-scale data or complex models such as large language models (LLM). A widely adopted strategy in post-training pruning is layer-wise reconstruction error minimization, which ensures that the pruned model maintains accuracy by preserving layer-wise output approximations (Frantar & Alistarh, 2023; Sun et al., 2024; Meng et al., 2024). This approach enables efficient pruning of large models by solving smaller subproblems independently for each layer, reducing computational overhead while preserving the signal in model representations. Optimization-based techniques have been proposed to refine this idea further, improving sparsity while minimizing performance degradation. Additionally, extensions of these methods explore structured sparsity patterns, such as block-wise sparsity, to enhance hardware efficiency while maintaining competitive accuracy. Also, recent studies hint at the possibility of finding an initial random sparse network that can be trained to achieve comparable performance to dense networks, although this generally involves several rounds of expensive training (Frankle & Carbin, 2019). Among various approaches, this work primarily focuses on pruning during training (Peste et al., 2021; Zhou et al., 2021; Kusupati et al., 2020; Lin et al., 2020; Sanh et al., 2020; Evci et al., 2020), which is known for achieving best results by guiding the model toward desired sparsity during training (Hoefler et al., 2021), and can be ideal for moderately sized models with adequate computational resources for training. Despite this, preserving the original dense performance remains challenging due to the complexity of tasks handled by deep learning models, often leading to the use of heuristics to manage these difficulties.

### 2.2. Flat Minima

In-depth empirical analyses into the optimization properties of deep neural network training, especially with regards to mini-batch training, have revealed a surprising correlation between well-generalizing solutions and their flatness (Keskar et al., 2017; Jiang et al., 2020b). This finding has prompted numerous studies aiming to understand the precise nature of this relationship (Andriushchenko et al., 2023; Neyshabur et al., 2017; Zhou et al., 2020), and its impact on neural network pruning, as explored by Lee et al. (2021), who link the challenge of training highly sparse networks to sharper loss landscapes based on an analysis of scaling properties under varying sizes of mini-batches and classical optimization theory.

This also motivated researchers to develop various techniques to explicitly induce flat minima during training (Foret et al., 2021; Izmailov et al., 2018; Orvieto et al., 2022; Chaudhari et al., 2017). Among them, sharpness-aware min-

imization (SAM) by Foret et al. (2021) aims to tackle this by solving the following min-max optimization problem:

$$\min_x \max_{\|\epsilon\|_2 \leq \rho} f(x + \epsilon), \qquad (2)$$

where we minimize the objective function over the entire $\epsilon$-neighborhood with radius $\rho$, *i.e.*, seek flat minima. Solving the inner maximization problem for the first-order Taylor approximation gives the following update rule for SAM:

$$x_{t+1} = x_t - \eta \nabla f \left( x_t + \rho \frac{\nabla f(x_t)}{\|\nabla f(x_t)\|_2} \right).$$

This has been shown to be effective in improving generalization performance and robustness across various domains (Chen et al., 2022; Bahri et al., 2022; Baek et al., 2024).

The success of sharpness minimization techniques has naturally led to exploring their implications for neural network pruning. Na et al. (2022) studied whether a flatter loss landscape can be more compressible by employing SAM during iterative magnitude pruning for fine-tuning BERT models. Shin et al. (2025) observed that the generalization benefits of SAM can be leveraged with sparsity for overparameterized neural networks. Inspired by SAM, Peste et al. (2022) proposed compression-aware minimization (CrAM) to minimize the loss increase induced by perturbation from compression, which aims to induce robustness to post-training one-shot pruning of any sparsity. Bair et al. (2023) suggested performing additional sharpness-aware training to pre-trained models, with larger perturbations given to coordinates of low importance score (*i.e.*, parameters to be pruned) to incur less loss increase when pruning.

While these efforts represent initial attempts to verify the effectiveness of sharpness minimization or its loosely inspired variants in enhancing pruning, we believe that sharpness minimization can be more effectively integrated into the sparsification process. Thus, in this work, we aim to weave this sharpness-minimization objective with the sparsification process to enhance the quality of the sparsified network via principled optimization-based approaches that are well established in the literature.

## 3. Method

In this section, we present a detailed derivation of our flatness-inducing sparsification algorithm S*parsification via* A*DMM with* F*latness* E*nforcement* or SAFE.

### 3.1. Problem Formulation

We begin by proposing the following min-max optimization problem with sparsity constraint:

$$\min_{\|x\|_0 \leq d} \max_{\|\epsilon\|_2 \leq \rho} f(x + \epsilon), \qquad (3)$$

where $f$ is the objective function to minimize, $d$ is the number of parameters to preserve, and $\rho$ is the radius of the perturbation $\epsilon$. Thus, the goal is to find a sparse solution $x^\star$ that minimizes the objective function in the whole $\epsilon$-neighborhood, *i.e.*, seek flat minima.

### 3.2. Augmented Lagrangian Based Approach

A standard approach for solving such constrained optimization problems is to employ Lagrangian duality or projected gradient descent. However, the discrete nature of $L_0$-norm makes Lagrangian duality infeasible, while projected gradient descent, despite its computational efficiency for $L_0$ constraints, can struggle with highly non-convex objectives in neural network optimization. To leverage the smooth optimization of Lagrangian and the efficiency of projection, we leverage augmented Lagrangian as described below.

To achieve this, we first employ variable splitting, a widely used trick, usually to separately deal with objective minimization and constraint satisfaction (Boyd et al., 2011). Precisely, instead of directly imposing the sparsity constraint on variable $x$, we first split it into variables $x$ and $z$ as follows:

$$\min_{x,z} \max_{\|\epsilon\|_2 \leq \rho} f(x + \epsilon) + I_{\|\cdot\|_0 \leq d}(z) \quad \text{s.t. } x = z,$$

where $I_{\|\cdot\|_0 \leq d}(z)$ is an indicator function for the sparsity constraint:

$$I_{\|\cdot\|_0 \leq d}(z) := \begin{cases} 0 & \text{if } \|z\|_0 \leq d \\ \infty & \text{else.} \end{cases}$$

We then slightly alter the Lagrangian by adding a penalty term $\lambda/2\|x-z\|_2^2$ with penalty parameter $\lambda$, which preserves equivalence to the original problem while also acting as a proximal term for the projection step. This alteration is a form of augmented Lagrangian, which we apply to form the Lagrangian dual problem of the following:

$$\max_u, \min_{x,z} \left( \mathcal{L}(x,z,u) := \max_{\|\epsilon\|_2 \leq \rho} f(x+\epsilon) + I_{\|\cdot\|_0 \leq d}(z) \right.$$
$$\left. - \frac{\lambda}{2}\|u\|_2^2 + \frac{\lambda}{2}\|x - z + u\|_2^2 \right),$$

where $u$ is a scaled dual variable for the equality constraint scaled by $1/\lambda$. Here, the projection can be computed efficiently via hard thresholding operation (Blumensath & Davies, 2009), which sets all entries of $x$ but the $d$ elements with the largest magnitudes to zero. Applying dual ascent leaves us with the following $x, z$-minimization and $u$-maximization:

$$x_{k+1}, z_{k+1} = \operatorname*{argmin}_{x,z} \max_{\|\epsilon\|_2 \leq \rho} f(x+\epsilon) + I_{\|\cdot\|_0 \leq d}(z)$$
$$+ \frac{\lambda}{2}\|x - z + u_k\|_2^2$$
$$u_{k+1} = \operatorname*{argmax}_u \frac{\lambda}{2}\|x_{k+1} - z_{k+1} + u\|_2^2 - \frac{\lambda}{2}\|u\|_2^2.$$

To minimize each $x$ and $z$ separately with iterative first-order optimization and exact projection operation respectively, we compute $x$ and $z$ in an alternating manner which gives the following iteration:

$$x_{k+1} = \arg\min_x \max_{\|\epsilon\|_2 \leq \rho} f(x + \epsilon) + \frac{\lambda}{2}\|x - z_k + u_k\|_2^2$$
$$z_{k+1} = \text{proj}_{\|\cdot\|_0 \leq d}(x_{k+1} + u_k)$$
$$u_{k+1} = u_k + x_{k+1} - z_{k+1},$$

where $\text{proj}_{\|\cdot\|_0 \leq d}$ is a projection operation onto the sparsity constraint (*i.e.*, the hard thresholding operator), and $u$-maximization is solved through applying a single step of gradient ascent with a step size of $\lambda$ on $y$, which is to ensure that the iterate stays within feasibility once reached.

### 3.3. $x$-minimization

For the $x$-minimization step, we first approximately solve the $\epsilon$-maximization via first-order approximation of $f$:

$$\epsilon^\star(x) \approx \arg\max_{\|\epsilon\|_2 \leq \rho} f(x) + \epsilon^\top \nabla f(x) = \rho \frac{\nabla f(x)}{\|\nabla f(x)\|_2},$$

which we apply back to the objective

$$x_{k+1} = \arg\min_x f(x + \epsilon^\star(x)) + \frac{\lambda}{2}\|x - z_k + u_k\|_2^2.$$

We solve this using gradient descent, where we remove higher-order terms in the gradient as in Foret et al. (2021),

$$\nabla_x \left( f(x + \epsilon^\star(x)) + \frac{\lambda}{2}\|x - z_k + u_k\|_2^2 \right)$$
$$= (I + \nabla\epsilon^\star(x))\nabla f(x)\big|_{x+\epsilon^\star(x)} + \lambda(x - z_k + u_k)$$
$$= \nabla f\left(x + \rho\frac{\nabla f(x)}{\|\nabla f(x)\|_2}\right) + \lambda(x - z_k + u_k), \quad (4)$$

thus leading to the following $x$-minimization steps:

$$x_k^{(t+1)} = x_k^{(t)} - \eta^{(t)}\left(\nabla f\left(x_k^{(t)} + \rho\frac{\nabla f(x_k^{(t)})}{\|\nabla f(x_k^{(t)})\|_2}\right)\right.$$
$$\left. + \lambda(x_k^{(t)} - z_k + u_k)\right), \quad (5)$$

where $t, \eta^{(t)}$ are the current step of $x$-minimization and its step-size, respectively.

### 3.4. Extension to Generalized Projection

While the Euclidean projection onto an $L_0$ constraint naturally yields magnitude-based sparsification, this often yields subpar performance in practice compared to more advanced saliency scores that account for the objective function. To naturally integrate these into the projection operation, we design a generalized distance metric that introduces a positive-definite diagonal matrix $\mathbf{P}$ that provides a framework to incorporate these advanced saliencies of the form $\mathbf{P}_{[i,i]}^{1/2}|x_{[i]}|$ in a principled manner:

$$z_{k+1} = \text{proj}_{\|\cdot\|_0 \leq d}^{\mathbf{P}}(x_{k+1} + u_k)$$
$$:= \arg\min_{\|z\|_0 \leq d} \frac{1}{2}\|z - (x_{k+1} + u_k)\|_{\mathbf{P}}^2$$
$$= \arg\min_{\|z\|_0 \leq d} \frac{1}{2}(z - (x_{k+1} + u_k))^\top \mathbf{P}(z - (x_{k+1} + u_k))$$

Geometrically, this can be understood as modifying the underlying distance metric to better represent the local geometric structure (*e.g.*, the Hessian) of the original objective function. We call this SAFE$^+$, where we leverage various saliency scores within the projection step, which we describe in detail below.

We lay out some notable examples of advanced saliency scores and the corresponding $\mathbf{P}$. The simplest case is $\mathbf{P} = \mathbf{I}$, where the projection reduces to standard hard thresholding as it corresponds to the Euclidean norm, yielding the original SAFE. Taking this further, setting it as the diagonal Hessian, *i.e.*, $\mathbf{P} = \text{diag}(\nabla^2 f(x))$, corresponds to Optimal Brain Damage (LeCun et al., 1989), a second-order pruning method that aims to remove parameters with minimal impact on the loss function. Also, using $\mathbf{P} = \text{diag}(\nabla f(x)\nabla f(x)^\top)$ aligns with the first-order pruning method SNIP (Lee et al., 2019), which removes parameters based on gradient sensitivity. Furthermore, Wanda (Sun et al., 2024), a layer-wise pruning method for language models, corresponds to taking $\mathbf{P} = \text{diag}(\mathbf{A}^\top\mathbf{A})$ where $\mathbf{A} \in \mathbb{R}^{N \times d}$ is an activation with batch size $N$ and feature dimension $d$ from a particular layer to prune. This corresponds to the diagonal Hessian of the reconstruction error for a single linear layer (Sun et al., 2024).

This generalized projection allows SAFE$^+$ to integrate diverse sparsification strategies all within its constrained optimization framework, thus enhancing both effectiveness and robustness in model pruning. Our empirical evaluation in Section 4.3 demonstrates its effectiveness in large language model pruning, though the methodology is not confined to this domain and is widely applicable.

### 3.5. Final Algorithm: SAFE and SAFE$^+$

Our final algorithm is summarized in Algorithm 1. We provide an intuitive description of how our algorithm performs sparsification. Every few steps of $x$-minimization, SAFE observes where the closest point on the sparsity constraint from the current $x$ is and registers it on $z$. While performing flatness-inducing minimization of the objective function on $x$, it penalizes the $x$ iterate to slightly move closer to $z$, the

**Algorithm 1** SAFE and SAFE$^+$ algorithms

---

**Require:** Target parameter count $d$, total train iteration $T$, dual-update interval $K$, learning rate $\eta^{(t)}$, perturbation radius $\rho$, penalty parameter $\lambda$, importance matrix $\mathbf{P}$.
 1: Initialize $x^{(0)}$
 2: $u = \mathbf{0}$
 3: **for** $t$ in $T$ **do**
 4:   **if** $t \mod K = 0$ **then**
 5:     **if** SAFE **then**
 6:       $z = \text{proj}_{\|\cdot\|_0 \leq d}(x^{(t+1)} + u)$
 7:     **else if** SAFE$^+$ **then**
 8:       $z = \text{proj}_{\|\cdot\|_0 \leq d}^{\mathbf{P}}(x^{(t+1)} + u)$
 9:     **end if**
10:     $u = u + x^{(t+1)} - z$
11:   **end if**
12:   $x^{(t+1/2)} = x^{(t)} - \eta^{(t)}\nabla f\left(x^{(t)} + \rho \cdot \frac{\nabla f(x^{(t)})}{\|\nabla f(x^{(t)})\|_2}\right)$
13:   $x^{(t+1)} = x^{(t+1/2)} - \eta^{(t)}\lambda(x^{(t)} - z + u)$
14: **end for**
15: **return** $\text{proj}_{\|\cdot\|_0 \leq d}(x^{(T)})$

---

latest estimate of the sparse solution. This gradually moves the dynamics of $x$ towards sparsity without incurring a sudden change of loss, all while performing flatness induction, yielding a sparse and flat minima.

In practice, particularly for image classification, we introduce scheduling to the penalty parameter $\lambda$ from zero to the target value in a cosine curve in order to apply less restriction in the initial phases of training, which slightly improves performance. Details of the ablation study on this scheduling strategy can be found in Appendix F.3.

### 3.6. Convergence Analysis

Here we present a convergence analysis of SAFE. Precisely, we first prove that our proposed iterative sharpness minimization in the $x$-update converges, then build the rest of the proof upon a well-studied result of ADMM (Boyd et al., 2011; Wang et al., 2019; Huang et al., 2021).

We start with standard assumptions used in the literature:

**Assumption 3.1.** (Lower bounded on constraint) The function $f$ is lower bounded on $\mathcal{A}$. That is, there exists a constant $f_{\min} := \min_{a \in \mathcal{A}} f(a)$ and $f_{\min} > -\infty$.

**Assumption 3.2.** ($\beta$-smoothness) The function $f$ is differentiable, and its gradient is $\beta$-smooth. That is, $\|\nabla f(x) - \nabla f(y)\| \leq \beta\|x - y\|$

**Assumption 3.3.** ($\mu$-weak convexity) There exists a constant $\mu \geq 0$ such that $f$ is $\mu$-weakly convex. i.e., $f(x) + \frac{\mu}{2}\|x\|^2$ is convex.

We also define the following notion of stationarity for the optimization problem (1) from Huang et al. (2021):

**Definition 3.4.** ($\delta$-stationary point) We say a point $\bar{x}$ is a $\delta$-stationary point of the optimization problem (1) if $\bar{x} \in \arg\min_{a \in \mathcal{A}} \|a - (\bar{x} - \delta^{-1}\nabla f(\bar{x}))\|$,

*i.e.*, the point $\bar{x}$ cannot be locally improved using projected gradient descent with step-size $\delta^{-1}$. With this definition, we ultimately demonstrate that SAFE converges to this $\delta$-stationary point, which is a necessary condition for the optimal solution to problem (1).

We first provide the central lemma on the convergence of our sharpness minimizing $x$ iterates:

**Lemma 3.5.** *(Convergence of $x$-minimization) Suppose that Assumptions 3.1 and 3.2 hold and let $\{x_k^{(t)}\}$ be generated by Equation (5) in Algorithm 1 with step-size $\eta^{(t)}$ and perturbation radius $\rho^{(t)}$ satisfying $\sum_{t=1}^{\infty} \eta^{(t)} = \infty, \sum_{t=1}^{\infty} \eta^{(t)}\rho^{(t)} < \infty, \limsup_t \rho^{(t)} < 1/\beta$. Let $\hat{\mathcal{L}}(x) = f(x) + \frac{\lambda}{2}\|x - z + u\|_2^2$ and assume that $\inf_{x \in \mathbb{N}} \hat{\mathcal{L}}(x^{(t)}) > -\infty$. Then $\nabla\hat{\mathcal{L}}(x^{(t)}) \to 0$.*

The detailed proof is provided in Appendix A.1. This shows that running Equation (5) produces a sequence that converges to the stationary point of the augmented Lagrangian $\hat{\mathcal{L}}$ with respect to $x$.

We use this to derive the convergence of SAFE as the following corollary:

**Corollary 3.6.** *(Convergence of SAFE) Suppose that Assumptions 3.1-3.3 hold. Assume further that $\delta$ is chosen large enough so that $\delta^{-1}\beta^2 - (\delta - \mu)/2 < 0$. Let $(\bar{x}, \bar{z}, \bar{u})$ be a limit point of SAFE algorithm. Then $\bar{x}$ is a $\delta$-stationary point of the optimization problem (1).*

This demonstrates that SAFE converges to the stationary point of the sparsity-constrained optimization problem (1). We note that, while the technical contributions of our analysis might be considered modest, SAFE is built on a theoretically rigorous foundation, unlike many other pruning techniques that often rely primarily on ad-hoc intuitions.

## 4. Experiments

In this section, we demonstrate that SAFE converges to sparse and flat solutions, leading to performance improvements over baselines in both image classification and language modeling tasks. We also show that SAFE is robust to noisy label training and corruptions during inference. The codes to reproduce the results are provided in `JAX` and `PyTorch`, with further details provided in Appendix B.5.

### 4.1. Convergence to Sparse and Flat Solutions

We first show that SAFE successfully guides training towards sparse and flat solutions compared to naive baselines on a simple neural network model. Specifically, we analyze the

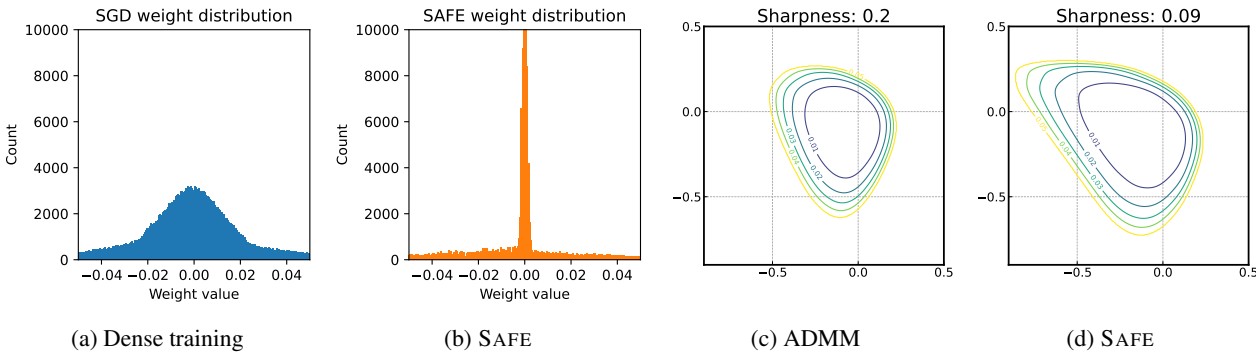

(a) Dense training      (b) SAFE      (c) ADMM      (d) SAFE

Figure 1: (a-b) Weight distributions of densely-trained model and model trained with SAFE, and (c-d) loss landscape and maximum Hessian eigenvalue of minima found by ADMM and SAFE. SAFE yields sparse and flat solutions.

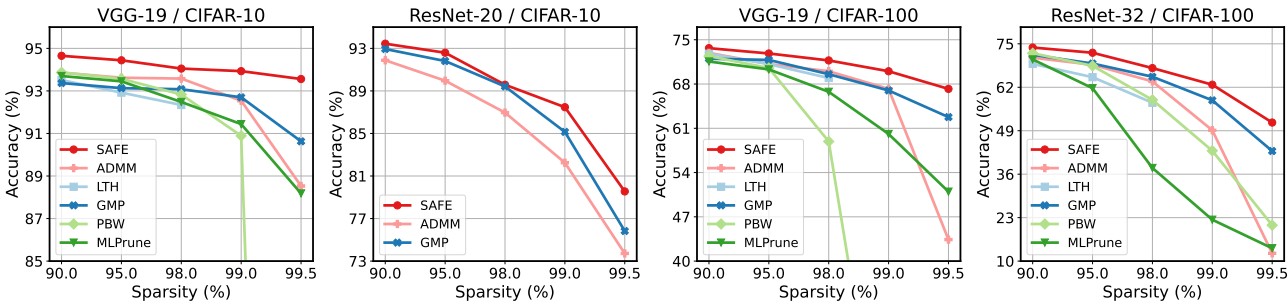

Figure 2: Validation accuracy (mean±std) of VGG-19 and ResNet-20/32 models on CIFAR-10/100 pruned across different sparsity levels and methods. SAFE consistently achieves superior performance across a broad range of sparsity levels.

weight distributions of models trained with standard dense training and SAFE to assess its sparsification capability. We also measure sharpnesses of SAFE and compare it to that of ADMM (Zhang et al., 2018) as a non-sharpness-minimizing baseline, by computing maximum Hessian eigenvalues and visualizing loss landscapes. The results are presented in Figure 1. Our findings indicate that SAFE effectively induces sparsity while simultaneously enforcing flatness, as evidenced by the concentration of weights near zero in contrast to dense training and a wider minimum with a lower Hessian eigenvalue compared to ADMM. This result demonstrates the effectiveness of SAFE in tackling the sharpness-aware sparsity-constrained optimization problem (3). Further experimental details are provided in Appendix B.2.

### 4.2. Evaluations on Image Classification

In this section, we show that SAFE can achieve outstanding generalization performance among various methods for CIFAR-10/100 image classification tasks (Krizhevsky et al., 2009). Specifically, we evaluate pruning performance on VGG-19 (Simonyan, 2014) and ResNet-20/32[1] (He et al.,

---

[1]Following the convention of Wang et al. (2020); Zhou et al. (2021), we double the number of channels from standard ResNet models.

2016) using a range of representative pruning methods, including PBW (Han et al., 2015), GMP (Kurtic & Alistarh, 2022; Zhu & Gupta, 2017), LTH (Liu et al., 2024; Frankle & Carbin, 2019), ADMM (Zhang et al., 2018), and MLPrune (Zeng & Urtasun, 2018), some of which achieves competitive to state-of-the-art performance in image classification (Hoefler et al., 2021). We mostly use standard values for common hyperparameters such as training epochs, learning rate, and weight decay (Zhou et al., 2021) and tune the hyperparameters unique to SAFE, which we report in detail in Appendix B. Notably, we do not perform additional training after pruning, and instead perform a cost-efficient statistical correction on the batch-norm layers with only a few forward passes (batch-norm tuning or BNT), which is a common practice in the literature (Hubara et al., 2021; Frantar & Alistarh, 2022; Peste et al., 2022). We refer to Appendix B.3 for full experimental details. The final validation accuracies are provided in Figure 2 and Table 7 of Appendix C.

Our findings show that across most configurations and sparsity levels, SAFE generally outperforms all baselines. Also, SAFE exhibits greater robustness under extreme sparsity compared to non-sharpness-minimized approaches (e.g., 99.5%). Crucially, SAFE achieves these results without requiring costly retraining, whereas PBW and LTH depend

Table 1: Perplexities (mean±std) of LLaMa models pruned to various sparsity levels using different methods. SAFE achieves competitive performance, while SAFE$^+$ outperforms baselines across all settings.

| | | LLaMa-2 | | | | LLaMa-3 | |
| | | 7B | | 13B | | 8B | |
| Sparsity | Method | Wikitext/C4 | | Wikitext/C4 | | Wikitext/C4 | |
|---|---|---|---|---|---|---|---|
| 0% | Dense | 5.47 | / 7.26 | 4.88 | / 6.72 | 6.23 | / 9.53 |
| 50% | Magnitude | 16.03 | / 21.33 | 6.82 | / 9.37 | 134.20 | / 273.3 |
| | SparseGPT | $6.99_{\pm0.03}$ | / $9.20_{\pm0.03}$ | $6.06_{\pm0.03}$ | / $8.20_{\pm0.01}$ | $9.36_{\pm0.11}$ | / $13.96_{\pm0.02}$ |
| | Wanda | $6.92_{\pm0.01}$ | / $9.23_{\pm0.00}$ | $5.98_{\pm0.01}$ | / $8.28_{\pm0.01}$ | $9.71_{\pm0.03}$ | / $14.88_{\pm0.04}$ |
| | ALPS | $6.87_{\pm0.01}$ | / $8.98_{\pm0.00}$ | $5.96_{\pm0.02}$ | / $8.09_{\pm0.04}$ | $\underline{9.05}_{\pm0.12}$ | / $\underline{13.40}_{\pm0.06}$ |
| | SAFE | $\underline{6.78}_{\pm0.01}$ | / $\underline{8.93}_{\pm0.00}$ | $\underline{5.76}_{\pm0.01}$ | / $\underline{7.85}_{\pm0.02}$ | $9.59_{\pm0.06}$ | / $14.60_{\pm0.04}$ |
| | SAFE$^+$ | $\mathbf{6.56}_{\pm0.01}$ | / $\mathbf{8.71}_{\pm0.00}$ | $\mathbf{5.67}_{\pm0.01}$ | / $\mathbf{7.74}_{\pm0.01}$ | $\mathbf{8.62}_{\pm0.06}$ | / $\mathbf{13.26}_{\pm0.06}$ |
| 60% | Magnitude | 1864 | / 2043 | 11.81 | / 14.62 | 5335 | / 7438 |
| | SparseGPT | $10.19_{\pm0.08}$ | / $12.86_{\pm0.05}$ | $8.31_{\pm0.09}$ | / $10.85_{\pm0.09}$ | $15.46_{\pm0.40}$ | / $21.25_{\pm0.18}$ |
| | Wanda | $10.75_{\pm0.07}$ | / $13.87_{\pm0.01}$ | $8.43_{\pm0.07}$ | / $11.55_{\pm0.01}$ | $22.06_{\pm0.19}$ | / $32.28_{\pm0.37}$ |
| | ALPS | $9.55_{\pm0.00}$ | / $\underline{11.24}_{\pm0.03}$ | $7.54_{\pm0.03}$ | / $9.87_{\pm0.05}$ | $\underline{14.03}_{\pm0.35}$ | / $\underline{18.72}_{\pm0.15}$ |
| | SAFE | $\underline{9.20}_{\pm0.04}$ | / $11.51_{\pm0.04}$ | $\underline{7.18}_{\pm0.03}$ | / $\underline{9.59}_{\pm0.03}$ | $15.90_{\pm0.25}$ | / $22.26_{\pm0.16}$ |
| | SAFE$^+$ | $\mathbf{8.30}_{\pm0.06}$ | / $\mathbf{10.59}_{\pm0.00}$ | $\mathbf{6.78}_{\pm0.04}$ | / $\mathbf{9.02}_{\pm0.15}$ | $\mathbf{12.18}_{\pm0.22}$ | / $\mathbf{17.30}_{\pm0.02}$ |
| 4:8 | Magnitude | 15.91 | / 31.61 | 7.32 | / 9.96 | 212.5 | / 336.3 |
| | SparseGPT | $8.42_{\pm0.05}$ | / $10.73_{\pm0.03}$ | $7.02_{\pm0.06}$ | / $9.33_{\pm0.04}$ | $12.16_{\pm0.20}$ | / $17.36_{\pm0.06}$ |
| | Wanda | $8.64_{\pm0.03}$ | / $11.35_{\pm0.01}$ | $7.01_{\pm0.02}$ | / $9.70_{\pm0.03}$ | $13.84_{\pm0.04}$ | / $21.14_{\pm0.06}$ |
| | ALPS | $\underline{8.11}_{\pm0.09}$ | / $\underline{10.21}_{\pm0.04}$ | $6.81_{\pm0.07}$ | / $9.33_{\pm0.04}$ | $\underline{11.38}_{\pm0.17}$ | / $\underline{16.10}_{\pm0.10}$ |
| | SAFE | $8.21_{\pm0.01}$ | / $10.61_{\pm0.04}$ | $6.60_{\pm0.02}$ | / $\underline{8.95}_{\pm0.02}$ | $12.15_{\pm0.14}$ | / $17.90_{\pm0.15}$ |
| | SAFE$^+$ | $\mathbf{7.59}_{\pm0.03}$ | / $\mathbf{9.88}_{\pm0.01}$ | $\mathbf{6.37}_{\pm0.03}$ | / $\mathbf{8.61}_{\pm0.01}$ | $\mathbf{10.51}_{\pm0.13}$ | / $\mathbf{15.67}_{\pm0.02}$ |
| 2:4 | Magnitude | 37.77 | / 74.70 | 8.88 | / 11.72 | 792.8 | / 2245 |
| | SparseGPT | $11.00_{\pm0.20}$ | / $13.54_{\pm0.03}$ | $8.78_{\pm0.09}$ | / $11.26_{\pm0.11}$ | $15.87_{\pm0.32}$ | / $22.45_{\pm0.12}$ |
| | Wanda | $12.17_{\pm0.02}$ | / $15.60_{\pm0.11}$ | $9.01_{\pm0.04}$ | / $12.40_{\pm0.01}$ | $23.03_{\pm0.38}$ | / $34.91_{\pm0.31}$ |
| | ALPS | $\underline{9.99}_{\pm0.19}$ | / $\underline{12.04}_{\pm0.04}$ | $8.16_{\pm0.17}$ | / $10.35_{\pm0.18}$ | $\underline{14.53}_{\pm0.33}$ | / $\underline{19.74}_{\pm0.18}$ |
| | SAFE | $10.53_{\pm0.13}$ | / $13.20_{\pm0.07}$ | $\underline{7.64}_{\pm0.05}$ | / $\underline{10.10}_{\pm0.01}$ | $17.49_{\pm0.27}$ | / $24.45_{\pm0.13}$ |
| | SAFE$^+$ | $\mathbf{8.96}_{\pm0.07}$ | / $\mathbf{11.34}_{\pm0.03}$ | $\mathbf{7.20}_{\pm0.04}$ | / $\mathbf{9.52}_{\pm0.01}$ | $\mathbf{13.39}_{\pm0.23}$ | / $\mathbf{19.03}_{\pm0.01}$ |

on multiple rounds of retraining. These results highlight the effectiveness of SAFE in preserving model accuracy during sparsification, especially under aggressive pruning scenarios.

### 4.3. Evaluation on Large Language Model Pruning

Here we scale our evaluations to modern large-scale settings and demonstrate that SAFE also delivers competitive performance against state-of-the-art LLM post-training pruning techniques.

For this purpose, we adapt SAFE and SAFE$^+$ to sequentially optimize the reconstruction error minimization (REM) objective for each transformer block (Shin et al., 2024), similarly to other LLM pruning techniques. For SAFE$^+$, we incorporate Wanda projection $z$-step, which identifies superior subnetworks compared to naive magnitude-based pruning in LLMs without compromising efficiency (Sun et al., 2024).

With this, we prune one of the most widely adopted language model family, LLaMA2-7b/13b (Touvron et al., 2023) and the more recent LLaMA3-8b (Meta, 2024), to 50% and 60% sparsities, as well as structured 4:8 and 2:4 sparsities. We compare SAFE with state-of-the-art LLM post-training

pruning methods such as SparseGPT (Frantar & Alistarh, 2023), Wanda (Sun et al., 2024), ALPS (ADMM-based) (Meng et al., 2024), as well as magnitude pruning (Han et al., 2015) and evaluate the perplexity on Wikitext2 (Merity et al., 2022) and C4 validation sets. We follow the common practice of randomly selecting 128 samples from the C4 training dataset (Raffel et al., 2020). We refer to Appendix B.4 for experimental details. The results are reported in Table 1.

We find that SAFE performs competitively to state-of-the-art methods, while SAFE$^+$ surpasses them across all models and sparsity settings. Considering that these baselines are tailored specifically to pruning LLMs, this demonstrates the flexibility of SAFE in adapting to different scenarios. Moreover, our method is more efficient than ALPS, which requires $\times 2.54$ more runtime than SAFE (see Appendix E.2).

### 4.4. Robustness to Noisy Data

Noisy data pose significant challenges in real-world scenarios. To address this, we evaluate SAFE on three representative challenges: incorrect training labels (Song et al., 2022), inference-time input corruption that arises naturally (Hendrycks & Dietterich, 2019), and corruptions that are de-

Table 2: Noisy label training. Validation accuracy is measured for sparse models trained with ADMM and SAFE under various levels of label noise and sparsity. SAFE is much more robust to label noise.

| Sparsity | Method | Noise ratio | | |
|---|---|---|---|---|
| | | 25% | 50% | 75% |
| 70% | ADMM | $77.00_{\pm0.91}$ | $59.18_{\pm0.55}$ | $32.62_{\pm0.89}$ |
| | SAFE | $\mathbf{90.58}_{\pm0.30}$ | $\mathbf{86.51}_{\pm0.16}$ | $\mathbf{67.01}_{\pm0.54}$ |
| 80% | ADMM | $76.18_{\pm0.56}$ | $62.67_{\pm0.38}$ | $32.86_{\pm1.12}$ |
| | SAFE | $\mathbf{91.25}_{\pm0.12}$ | $\mathbf{86.55}_{\pm0.07}$ | $\mathbf{66.49}_{\pm0.56}$ |
| 90% | ADMM | $79.40_{\pm0.12}$ | $66.64_{\pm0.13}$ | $36.84_{\pm0.94}$ |
| | SAFE | $\mathbf{90.68}_{\pm0.21}$ | $\mathbf{86.49}_{\pm0.06}$ | $\mathbf{64.72}_{\pm0.61}$ |
| 95% | ADMM | $77.71_{\pm0.52}$ | $67.10_{\pm1.37}$ | $39.68_{\pm1.44}$ |
| | SAFE | $\mathbf{89.86}_{\pm0.11}$ | $\mathbf{85.18}_{\pm0.15}$ | $\mathbf{64.25}_{\pm0.36}$ |

liberately introduced by adversaries (Szegedy et al., 2014).

**Training on noisy labels** To assess the robustness of SAFE against label noise, we randomly corrupt $\{25\%, 50\%, 75\%\}$ of labels in CIFAR-10 and use it to train ResNet-20 with both SAFE and ADMM. The same hyperparameters from Section 4.2 are used in all experiments. As presented in Table 2, we observe that SAFE consistently outperforms ADMM across all levels of label noise and sparsity, with accuracy improvements ranging from $+10\%$ to $+30\%$.

Additionally, we observe that ADMM relies heavily on sparsity to mitigate label noise, exhibiting an overall trend of increasing accuracy with higher sparsity levels. This dependence is further reflected in the sparse double descent pattern reported at the 25% noise ratio (He et al., 2022), where accuracy initially declines up to 80% sparsity, rises sharply to 79% at 90% sparsity, and then drops again to 77% at 95% sparsity. This contributes to the overall decreasing performance gap between ADMM and SAFE, which may be interpreted as the benefit of sharpness-minimization diminishing with fewer parameters (Shin et al., 2025). However, more crucially, the overall under-performance of ADMM indicates that sparsity alone is a poor remedy for label noise, highlighting the effectiveness of SAFE—particularly its flatness enforcement—in reducing the impact of label noise. This aligns well with previous observations that sharpness-minimization can enhance robustness toward label noise (Baek et al., 2024). Also, the lack of double descent in SAFE suggests that its effectiveness may be attributed to sharpness minimization functioning as an effective regularizer, as supported by the claims of Nakkiran et al. (2021) that 'optimal' regularization can mitigate the double descent phenomenon.

**Evaluation on corrupted image** We evaluate the sparse models trained with ADMM and SAFE, as obtained in Section 4.2, on the CIFAR-10 test set with common image corruptions and adversarial perturbations. Specifically, for

Table 3: Evaluation on corrupted data. CIFAR-10C is used for common corruptions, and $l_\infty$ and $l_2$ PGD attacks are used to generate adversarial corruption on the validation set of CIFAR-10. SAFE improves robustness over naturally and adversarially corrupted images.

| Sparsity | Method | Common corruption (avg.) | | Adversarial | |
|---|---|---|---|---|---|
| | | intensity=3 | intensity=5 | $l_\infty$-PGD | $l_2$-PGD |
| 90% | ADMM | $70.06_{\pm0.03}$ | $52.01_{\pm0.38}$ | $49.81_{\pm1.02}$ | $49.71_{\pm1.06}$ |
| | SAFE | $\mathbf{73.98}_{\pm0.09}$ | $\mathbf{55.11}_{\pm0.27}$ | $\mathbf{56.43}_{\pm1.03}$ | $\mathbf{56.36}_{\pm1.11}$ |
| 95% | ADMM | $68.87_{\pm0.25}$ | $50.56_{\pm0.07}$ | $49.84_{\pm1.78}$ | $49.68_{\pm1.79}$ |
| | SAFE | $\mathbf{72.92}_{\pm0.41}$ | $\mathbf{54.86}_{\pm0.51}$ | $\mathbf{51.40}_{\pm0.89}$ | $\mathbf{51.36}_{\pm0.94}$ |
| 98% | ADMM | $65.46_{\pm0.24}$ | $48.65_{\pm0.04}$ | $43.33_{\pm1.59}$ | $\mathbf{43.42}_{\pm1.60}$ |
| | SAFE | $\mathbf{68.20}_{\pm0.47}$ | $\mathbf{49.96}_{\pm0.83}$ | $\mathbf{43.34}_{\pm0.90}$ | $43.41_{\pm1.03}$ |
| 99% | ADMM | $59.21_{\pm0.47}$ | $43.81_{\pm0.44}$ | $30.29_{\pm0.64}$ | $30.32_{\pm0.58}$ |
| | SAFE | $\mathbf{66.02}_{\pm0.56}$ | $\mathbf{49.34}_{\pm1.03}$ | $\mathbf{43.70}_{\pm1.28}$ | $\mathbf{32.70}_{\pm1.28}$ |
| 99.5% | ADMM | $55.72_{\pm0.44}$ | $41.55_{\pm0.78}$ | $23.25_{\pm1.92}$ | $23.25_{\pm1.85}$ |
| | SAFE | $\mathbf{56.58}_{\pm0.36}$ | $\mathbf{42.27}_{\pm0.63}$ | $\mathbf{29.48}_{\pm0.68}$ | $\mathbf{29.45}_{\pm0.74}$ |

common corruptions, we use CIFAR-10C (Hendrycks & Dietterich, 2019), a benchmark consisting of CIFAR-10 test images corrupted with 19 types of real-world noise (*e.g.*, fog, snow, *etc.*) and distortions (*e.g.*, jpeg compression, contrast, *etc.*) at five levels of intensity. We average the performance across all corruption types for intensity levels 3 and 5. For adversarial noise, we follow Zhang et al. (2024) and use a 10-step Projected Gradient Descent (PGD) attack on each sparse model under $l_\infty$ and $l_2$ norm with bound $\epsilon = 1/255$ and $3/255$ and step size $\alpha = \epsilon/4$, respectively. As shown in Table 3, SAFE enhances robustness to both common and adversarial image corruptions, aligning with previous work on sharpness minimization (Zhang et al., 2024; Wei et al., 2023).

### 4.5. Comparison with Other SAM-based pruner

To strengthen the comparison with closely related baselines, we compare SAFE to two pruning baselines inspired by SAM—IMP+SAM (Na et al., 2022) and CrAM (Peste et al., 2022)—on ResNet-20/CIFAR-10 across multiple sparsity levels.

IMP+SAM (Na et al., 2022) involves applying SAM during iterative magnitude pruning (Liu et al., 2024; Frankle & Carbin, 2019). While it was initially introduced for language model finetuning, we adapt this method to image classification and use the same training epochs and sharpness-minimization hyperparameter search range as SAFE to ensure fair comparison. Pruning is performed every 10 epochs with sparsity increasing either linearly, following Na et al. (2022), or cubically (Zhu & Gupta, 2017), with the latter yielding better performance.

Compression-Aware Minimizer (CrAM) (Peste et al., 2022), on the other hand, extends upon the robust optimization principles of SAM to train compressible models by en-

Table 4: Comparison with IMP+SAM, CrAM, and CrAM$^+$ on ResNet-20/CIFAR-10. SAFE$_{+SG}$, which extends SAFE using a similar technique as CrAM$^+$, outperforms most baselines at moderate sparsity and all baselines at extreme sparsity.

| Method | Sparsity | | | |
|---|---|---|---|---|
| | 95% | 98% | 99% | 99.5% |
| IMP+SAM$_{linear}$ | $80.30_{\pm 0.12}$ | $36.03_{\pm 4.19}$ | $18.30_{\pm 2.80}$ | $13.80_{\pm 0.52}$ |
| IMP+SAM$_{cubic}$ | $92.50_{\pm 0.05}$ | $89.24_{\pm 0.06}$ | $83.74_{\pm 0.14}$ | $73.73_{\pm 0.30}$ |
| CrAM | $90.18_{\pm 1.80}$ | $69.53_{\pm 12.36}$ | $45.17_{\pm 20.86}$ | $10.00_{\pm 0.00}$ |
| CrAM$^+$ | $\mathbf{93.62}_{\pm 0.06}$ | $\mathbf{91.75}_{\pm 0.41}$ | $\underline{88.82}_{\pm 0.18}$ | $\underline{81.30}_{\pm 0.56}$ |
| SAFE | $\underline{92.59}_{\pm 0.09}$ | $89.58_{\pm 0.10}$ | $87.47_{\pm 0.07}$ | $79.55_{\pm 0.13}$ |
| SAFE$_{+SG}$ | $92.40_{\pm 0.06}$ | $\underline{90.09}_{\pm 0.13}$ | $\mathbf{89.13}_{\pm 0.06}$ | $\mathbf{85.85}_{\pm 0.09}$ |

couraging the models to maintain strong post-compression performance under the presence of small perturbations as $\min_x \max_{\|\epsilon\| \leq \rho} f(C(x + \epsilon))$ given some compression operation $C$. This leads to the CrAM update rule $x_{t+1} = x_t - \eta \nabla f(C(x + \rho \nabla f(x)))$. Along with this, we additionally compare with CrAM$^+$, a variant introduced by Peste et al. (2022) that simply adds the original gradient $\nabla f(x)$ in the update as $x_{t+1} = x_t - \eta [\nabla f(C(x + \rho \nabla f(x))) + \nabla f(x))]$. We run these using the optimal hyperparameters as suggested in Peste et al. (2022). It should be noted that the additional technique of CrAM$^+$ is somewhat auxiliary to the core robust optimization mechanism that connects CrAM to SAFE, and thus, care must be taken when associating their performance gains with the central strategy that defines CrAM. For better comparison, we extend this strategy to SAFE by adding the gradient computed at projected point $\nabla f(C(x))$ during iterative $x$-minimization of SAFE, which we denote as SAFE$_{+SG}$.

As shown in Table 4, SAFE and SAFE$_{+SG}$ outperforms IMP+SAM and CrAM, where SAFE$_{+SG}$ demonstrates competitive performance to CrAM$^+$ on moderate sparsity and outperforms it in extreme sparsity. We suspect that the pruning operation in IMP+SAM may be overly abrupt, potentially hindering the benefits of sharpness minimization. We additionally observe similarly strong performance over IMP+SAM on language model pruning in Appendix D.2. Conversely, although CrAM$^+$ achieves competitive results, its benefits fails to extend to extreme sparsity. More crucially, while SAFE achieves reliable performance without much additional techniques, CrAM, by itself, performs poorly in all sparsities, depending heavily on auxiliary techniques to drastically improve performance. This suggests that caution is warranted when attributing these gains to the effectiveness of their robust optimization formulation inspired by sharpness-minimization, which is more likely provided through the use of both the original dense gradient and the sparse gradient computed at the projected point

from CrAM$^+$. A similar trend is observed in SAFE$_{+SG}$, further supporting this interpretation. SAFE, on the other hand, delivers competitive performance without relying on these additional techniques, highlighting the intrinsic effectiveness of its smooth penalization via the augmented Lagrangian and split-variable structure of the ADMM framework to jointly balance sharpness minimization and sparsity. We provide additional comparison with other variants of CrAM in Appendix D.1.

## 5. Conclusion

In this work, we propose an effective and principled approach called SAFE to obtain sparse and flat solutions by solving a constrained optimization problem based on the augmented Lagrangian, which we further extend to SAFE$^+$ by proposing a generalization for the projection operation. We show that SAFE can be applied to neural network pruning, and as a result, it not only obtains the desired flatness as well as high sparsity in the given deep model, but also enhances its generalization performance quite significantly, far better than the compared baselines, as validated across standard benchmarks. Interestingly, SAFE preserves its robustness to various data noise during both training and inference, which stems from the original sharpness minimization strategy. Finally, we compare with more directly related SAM-inspired baselines, demonstrating the intrinsic effectiveness of SAFE without much reliance on auxiliary techniques. We believe that this principled approach for obtaining sparse and flat solutions—concepts that have often been explored rather separately in the literature—offers significant potential.

## Acknowledgements

This work was partly supported by the Institute of Information & communications Technology Planning & Evaluation (IITP) grant funded by the Korean government (MSIT) (IITP-2019-0-01906, Artificial Intelligence Graduate School Program (POSTECH) and RS-2022-II220959, (part2) Few-Shot learning of Causal Inference in Vision and Language for Decision Making), the National Research Foundation of Korea (NRF) grant funded by the Korean government (MSIT) (2022R1F1A1064569, RS-2023-00210466, RS-2023-00265444).

## Impact Statement

This paper presents work aimed at advancing the field of Machine Learning, with the potential to influence both theoretical understanding and practical applications. While our contributions do not directly raise immediate concerns requiring specific emphasis, we acknowledge that advancements in this domain can have far-reaching societal impli-

cations. We will ensure ongoing discourse on the broader impact of our work in diverse contexts if the need is later recognized.

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

# A. Convergence analysis of SAFE

In this section, we show that SAFE converges to a stationary point of the augmented Lagrangian. Precisely, we first prove that our proposed sharpness minimized $x$-minimization iterate converges to the stationary point for the augmented Lagrangian for the original loss function (*i.e.*, $\hat{\mathcal{L}}(x) = f(x) + \lambda/2\|x - z + u\|^2$) with respect to $x$. With this result, build the rest of the proof upon a well-studied result of ADMM (Boyd et al., 2011; Wang et al., 2019; Huang et al., 2021).

Prior to providing detailed proofs of core lemmas and corollaries, we first describe the properties of the augmented Lagrangian $\hat{\mathcal{L}}(x)$ based on the Assumptions 3.2 and 3.3.

Defining strong convexity as:

**Definition A.1. ($\alpha$-strong convexity)** Let $f$ be differentiable. $f$ is $\alpha$-strongly convex if there exists $\alpha > 0$ such that
$f(y) \geq f(x) + \langle \nabla f(x), y - x \rangle + \frac{\alpha}{2}\|y - x\|^2,$

we present the smoothness and strong convexity of $\hat{\mathcal{L}}(x)$ through the following lemma:

**Lemma A.2.** *Let $f$ be $\beta$-smooth and $\mu$-weakly convex. Then $\hat{\mathcal{L}}(x) = f(x) + \frac{\lambda}{2}\|x - z + u\|^2$ is also $(\beta + \lambda)$-smooth and $(\lambda - \mu)$-strongly convex for $\lambda > \mu$.*

*Proof.* From Theorem 3.2, we have

$$\|\nabla\hat{\mathcal{L}}(x) - \nabla\hat{\mathcal{L}}(x)\| = \|\nabla f(x) + \lambda(x - z + u) - \nabla f(y) - \lambda(y - z + u)\|$$
$$\leq \|\nabla f(x) - \nabla f(y)\| + \|\lambda x - \lambda y\|$$
$$\leq (\beta + \lambda)\|x - y\|$$

thus by definition, $\hat{\mathcal{L}}$ is $(\beta + \lambda)$-smooth.

Also, from first-order condition of convexity of $f(x) + \frac{\mu}{2}\|x\|^2$ from Theorem 3.3, we have

$$f(y) + \frac{\mu}{2}\|y\|^2 \geq f(x) + \frac{\mu}{2}\|x\|^2 + \langle \nabla f(x) + \mu x, y - x \rangle$$
$$\Rightarrow f(y) \geq f(x) + \langle \nabla f(x) + \mu x, y - x \rangle + \frac{\mu}{2}\|x\|^2 - \frac{\mu}{2}\|y\|^2$$
$$\Rightarrow f(y) \geq f(x) + \langle \nabla f(x) + \mu x, y - x \rangle - \frac{\mu}{2}\langle y + x, y - x \rangle$$
$$\Rightarrow \hat{\mathcal{L}}(y) \geq \hat{\mathcal{L}}(x) + \langle \nabla\hat{\mathcal{L}}(x), y - x \rangle - \frac{\mu}{2}\langle y - x, y - x \rangle + \frac{\lambda}{2}\langle y - x, y - x \rangle$$
$$\Rightarrow \hat{\mathcal{L}}(y) \geq \hat{\mathcal{L}}(x) + \langle \nabla\hat{\mathcal{L}}(x), y - x \rangle + \frac{\lambda - \mu}{2}\|y - x\|^2.$$

Since $\lambda - \mu > 0$, by definition $\hat{\mathcal{L}}$ is $(\lambda - \mu)$-strongly convex. $\qquad\square$

## A.1. Proof of Theorem 3.5

We provide the convergence proof of our sharpness minimizing $x$ iterates. We mostly follow the procedure of Khanh et al. (2024), with the objective and the update rule altered to match our configurations. Before we proceed, we recall prior results from Khanh et al. (2024)

**Lemma A.3.** *(**Lemma B.1** of Khanh et al. (2024)) Let $\{a^{(t)}\}, \{b^{(t)}\}, \{c^{(t)}\}$ be sequences of nonnegative numbers satisfying the conditions*

$$a^{(t+1)} - a^{(t)} \leq b^{(t)} a^{(t)} + c^{(t)} \text{ for sufficient large } t \in \mathbb{N}, \qquad (a)$$

$$\{b^{(t)}\} \text{ is bounded}, \sum_{t=1}^{\infty} b^{(t)} = \infty, \sum_{t=1}^{\infty} c^{(t)} < \infty, \text{ and } \sum_{t=1}^{\infty} b^{(t)} a^{(t)} < \infty. \qquad (b)$$

*Then we have that $a^{(t)} \to 0$ as $t \to \infty$*

With this, we derive the convergence of our sharpness minimizing $x$ iterates in the following lemma:

**Lemma A.4.** *(**Convergence of $x$-minimization**) Suppose that Theorem 3.1 and 3.2 hold and let $\{x_k^{(t)}\}$ be generated by Equation (5) in Algorithm 1 with step-size $\eta^{(t)}$ and perturbation radius $\rho^{(t)}$ satisfying*

$$\sum_{t=1}^{\infty} \eta^{(t)} = \infty, \sum_{t=1}^{\infty} \eta^{(t)} \rho^{(t)} < \infty, \limsup_t \rho^{(t)} < 1/\beta. \tag{6}$$

*Let $\hat{\mathcal{L}}(x) = f(x) + \frac{\lambda}{2}\|x - z + u\|_2^2$ and assume that $\inf_{x \in \mathbb{N}} \hat{\mathcal{L}}(x^{(t)}) > -\infty$. Then $\nabla\hat{\mathcal{L}}(x^{(t)}) \to 0$.*

*Proof.* Let the gradient of our sharpness minimizing $x$ iterate Equation (4) as

$$g^{(t)} := \nabla f\left(x^{(t)} + \rho^{(t)}\frac{\nabla f(x^{(t)})}{\|\nabla f(x^{(t)})\|}\right) + \lambda(x^{(t)} - z + u),$$

where we drop the subscript denoting the outer SAFE iterate for simplicity. We also denote $\hat{\beta} := \beta - \mu$. We first begin from the $\hat{\beta}$-smooothness of $\hat{\mathcal{L}}$ from Theorem A.2 as

$$
\begin{aligned}
\hat{\mathcal{L}}(x^{(t+1)}) &\leq \hat{\mathcal{L}}(x^{(t)}) + \langle \nabla\hat{\mathcal{L}}(x^{(t)}), x^{(t+1)} - x^{(t)}\rangle + \frac{\hat{\beta}}{2}\|x^{(t+1)} - x^{(t)}\|^2 \\
&= \hat{\mathcal{L}}(x^{(t)}) - \eta^{(t)}\langle \nabla\hat{\mathcal{L}}(x^{(t)}), g^{(t)}\rangle + \frac{\hat{\beta}\eta^{(t)2}}{2}\|g^{(t)}\|^2 \\
&= \hat{\mathcal{L}}(x^{(t)}) - \eta^{(t)}(1 - \hat{\beta}\eta^{(t)})\langle \nabla\hat{\mathcal{L}}(x^{(t)}), g^{(t)}\rangle + \frac{\hat{\beta}\eta^{(t)2}}{2}\left(\|g^{(t)} - \nabla\hat{\mathcal{L}}(x^{(t)})\|^2 - \|\nabla\hat{\mathcal{L}}(x^{(t)})\|^2\right). \tag{7}
\end{aligned}
$$

Here, we bound each $\|g^{(t)} - \nabla\hat{\mathcal{L}}(x^{(t)})\|$ and $\langle g^{(t)}, \nabla\hat{\mathcal{L}}(x^{(t)})\rangle$ using the $\hat{\beta}$-smoothness as follows

$$
\begin{aligned}
\|g^{(t)} - \nabla\hat{\mathcal{L}}(x^{(t)})\| &= \left\|\nabla f\left(x^{(t)} + \rho^{(t)}\frac{\nabla f(x^{(t)})}{\|\nabla f(x^{(t)})\|}\right) + \lambda(x^{(t)} - z + u) - \left(f(x^{(t)}) + \lambda(x^{(t)} - z + u)\right)\right\| \\
&= \left\|\nabla f\left(x^{(t)} + \rho^{(t)}\frac{\nabla f(x^{(t)})}{\|\nabla f(x^{(t)})\|}\right) - f(x^{(t)})\right\| \\
&\leq \beta\left\|x^{(t)} + \rho^{(t)}\frac{\nabla f(x^{(t)})}{\|\nabla f(x^{(t)})\|} - x^{(t)}\right\| \\
&= \beta\rho^{(t)}, \tag{8}
\end{aligned}
$$

and using this result, we have that

$$
\begin{aligned}
\langle g^{(t)}, \nabla\hat{\mathcal{L}}(x^{(t)})\rangle &= \langle g^{(t)} - \nabla\hat{\mathcal{L}}(x^{(t)}), \nabla\hat{\mathcal{L}}(x^{(t)})\rangle + \|\hat{\mathcal{L}}(x^{(t)})\|^2 \\
&\geq -\|g^{(t)} - \nabla\hat{\mathcal{L}}(x^{(t)})\| \cdot \|\nabla\hat{\mathcal{L}}(x^{(t)})\| + \|\hat{\mathcal{L}}(x^{(t)})\|^2 \\
&\geq -\beta\rho^{(t)}\|\nabla\hat{\mathcal{L}}(x^{(t)})\| + \|\hat{\mathcal{L}}(x^{(t)})\|^2. \tag{9}
\end{aligned}
$$

Applying Equation (8) and (9) back to Equation (7) gives

$$\hat{\mathcal{L}}(x^{(t+1)}) = \hat{\mathcal{L}}(x^{(t)}) - \eta^{(t)}(1 - \hat{\beta}\eta^{(t)})\langle\nabla\hat{\mathcal{L}}(x^{(t)}), g^{(t)}\rangle + \frac{\hat{\beta}\eta^{(t)2}}{2}\left(\|g^{(t)} - \nabla\hat{\mathcal{L}}(x^{(t)})\|^2 - \|\nabla\hat{\mathcal{L}}(x^{(t)})\|^2\right)$$

$$\leq \hat{\mathcal{L}}(x^{(t)}) - \eta^{(t)}(1 - \hat{\beta}\eta^{(t)})\left(-\hat{\beta}\rho^{(t)}\|\nabla\hat{\mathcal{L}}(x^{(t)})\| + \|\nabla\hat{\mathcal{L}}(x^{(t)})\|^2\right) + \frac{\hat{\beta}^3\eta^{(t)2}\rho^{(t)2}}{2} - \frac{\hat{\beta}\eta^{(t)2}}{2}\|\nabla\hat{\mathcal{L}}(x^{(t)})\|^2$$

$$= \hat{\mathcal{L}}(x^{(t)}) - \frac{\eta^{(t)}}{2}(2 - \hat{\beta}\eta^{(t)})\|\nabla\hat{\mathcal{L}}(x^{(t)})\|^2 + \hat{\beta}\eta^{(t)}\rho^{(t)}(1 - \hat{\beta}\eta^{(t)})\|\nabla\hat{\mathcal{L}}(x^{(t)})\| + \frac{\hat{\beta}^3\eta^{(t)2}\rho^{(t)2}}{2}$$

$$\leq \hat{\mathcal{L}}(x^{(t)}) - \frac{\eta^{(t)}}{2}(2 - \hat{\beta}\eta^{(t)})\|\nabla\hat{\mathcal{L}}(x^{(t)})\|^2 + \frac{\hat{\beta}\eta^{(t)}\rho^{(t)}}{2}(1 - \hat{\beta}\eta^{(t)})\left(1 + \|\nabla\hat{\mathcal{L}}(x^{(t)})\|^2\right) + \frac{\hat{\beta}^3\eta^{(t)2}\rho^{(t)2}}{2}$$

$$= \hat{\mathcal{L}}(x^{(t)}) - \frac{\eta^{(t)}}{2}(2 - \hat{\beta}\eta^{(t)} - \hat{\beta}\rho^{(t)} + \hat{\beta}^2\eta^{(t)}\rho^{(t)})\|\nabla\hat{\mathcal{L}}(x^{(t)})\|^2 + \frac{\hat{\beta}\eta^{(t)}\rho^{(t)}}{2}(1 - \hat{\beta}\eta^{(t)}) + \frac{\hat{\beta}^3\eta^{(t)2}\rho^{(t)2}}{2}$$

$$= \hat{\mathcal{L}}(x^{(t)}) - \frac{\eta^{(t)}}{2}(2 - \hat{\beta}\eta^{(t)} - \hat{\beta}\rho^{(t)} + \hat{\beta}^2\eta^{(t)}\rho^{(t)})\|\nabla\hat{\mathcal{L}}(x^{(t)})\|^2 + \hat{\beta}\eta^{(t)}\rho^{(t)}\left(\frac{1 - \hat{\beta}\eta^{(t)} + \hat{\beta}^2\eta^{(t)}\rho^{(t)}}{2}\right)$$

From here, we find $c_1 > 0$ and $c_2 \in (0, 1)$ and $T \in \mathbb{N}$ such that

$$\frac{1}{2}(2 - \hat{\beta}\eta^{(t)} - \hat{\beta}\rho^{(t)} + \hat{\beta}^2\eta^{(t)}\rho^{(t)}) \geq c_1, \quad \frac{1 - \hat{\beta}\eta^{(t)} + \hat{\beta}^2\eta^{(t)}\rho^{(t)}}{2} \leq c_2, \text{ and } \beta\eta^{(t)} < 1 \text{ for all } t > T,$$

where applying this gives us

$$\hat{\mathcal{L}}(x^{(t+1)}) \leq \hat{\mathcal{L}}(x^{(t)}) - c_1\eta^{(t)}\|\nabla\hat{\mathcal{L}}(x^{(t)})\|^2 + c_2\beta\eta^{(t)}\rho^{(t)}. \tag{10}$$

Also, defining $w^{(t)} := c_2\sum_{i=t}^{\infty}\beta\eta^{(i)}\rho^{(i)}$ for $t \in \mathbb{N}$, we get that $w^{(t)} \to 0$ as $t \to \infty$ and $w^{(t)} - w^{(t+1)} = c_2\beta\eta^{(t)}\rho^{(t)}$ for all $t \in \mathbb{N}$. Then Equation (10) can be rewritten as

$$\hat{\mathcal{L}}(x^{(t+1)}) + w^{(t+1)} \leq \hat{\mathcal{L}}(x^{(t)}) + w^{(t)} - c_1\eta^{(t)}\|\nabla\hat{\mathcal{L}}(x^{(t)})\|^2. \tag{11}$$

Here we telescope this bound from $t = T$ to $\infty$ and combine with $\inf f(x^{(t)}) > -\infty$ and $w^{(t)} \to 0$ as $t \to \infty$ that

$$c_1\sum_{t=T}^{\infty}\eta^{(t)}\|\nabla\hat{\mathcal{L}}(x^{(t)})\|^2 \leq \sum_{t=T}^{\infty}(\hat{\mathcal{L}}(x^{(t)}) - \hat{\mathcal{L}}(x^{(t+1)}) + w^{(t)} - w^{(t+1)}) \tag{12}$$

$$\leq \hat{\mathcal{L}}(x^{(K)}) - \inf_{T\in\mathbb{N}}\hat{\mathcal{L}}(x^{(t)}) + w^{(K)} < \infty \tag{13}$$

We finally employ Theorem A.3 with $a^{(t)} := \|\nabla\hat{\mathcal{L}}(x^{(t)})\|$, $b^{(t)} := \hat{\beta}\eta^{(t)}$, and $c^{(t)} := \hat{\beta}\eta^{(t)}\rho^{(t)}$ for all $t \in \mathbb{N}$ to derive $\hat{\mathcal{L}}(x^{(t)}) \to 0$. Here, condition (a) is satisfied due to the estimates

$$a^{(t+1)} - a^{(t)} = \|\nabla\hat{\mathcal{L}}(x^{(t+1)})\| - \|\nabla\hat{\mathcal{L}}(x^{(t)})\|$$

$$\leq \|\nabla\hat{\mathcal{L}}(x^{(t+1)}) - \nabla\hat{\mathcal{L}}(x^{(t)})\|$$

$$\leq \hat{\beta}\|x^{(t+1)} - x^{(t)}\| = \hat{\beta}\eta^{(t)}\|g^{(t)}$$

$$\leq \hat{\beta}\eta^{(t)}(\|\nabla\hat{\mathcal{L}}(x^{(t)})\| + \|g^{(t)} - \nabla\hat{\mathcal{L}}(x^{(t)})\|)$$

$$\leq \hat{\beta}\eta^{(t)}(\|\nabla\hat{\mathcal{L}}(x^{(t)})\| + \|g^{(t)} - \nabla\hat{\mathcal{L}}(x^{(t)})\|)$$

$$= b^{(t)}a^{(t)} + c^{(t)} \text{ for all } k \in \mathbb{N}.$$

Also, the conditions in (b) hold by Equation (6) and $\sum_{t=1}^{\infty}\eta^{(t)}\|\nabla\hat{\mathcal{L}}(x^{(t)})\|^2 < \infty$. Thus from Theorem A.3, $\|\nabla\hat{\mathcal{L}}(x^{(t)})\| = a^{(t)} \to 0$ as $t \to \infty$. $\qquad\square$

This shows that running Equation (5) produces a sequence that converges to the stationary point of the augmented Lagrangian $\hat{\mathcal{L}}(x, z, u) = f(x) + I_{\|\cdot\|_0 \leq d}(z) + \frac{\lambda}{2}\|u\|_2^2 + \frac{\lambda}{2}\|x - z + u\|_2^2$ with respect to $x$. This is crucial for the convergence of SAFE as we show in the following section.

## A.2. Proof of Theorem 3.6

Here we prove the convergence of SAFE. This is a straightforward procedure: given our convergence guarantee of the $x$ iterates in Theorem A.4, the convergence properties of SAFE can be described in terms of classical ADMM. Thus, in this section, we walk through the convergence analysis of ADMM as provided in Huang et al. (2021) and demonstrate how our sharpness minimizing $x$ iterates is applied within the proof.

**Corollary A.5.** *(Convergence of SAFE) Suppose that Assumptions 3.1-3.3 hold. Assume further that $\delta$ is chosen large enough so that $\delta^{-1}\beta^2 - (\delta - \mu)/2 < 0$. Let $(\bar{x}, \bar{z}, \bar{u})$ be a limit point of SAFE algorithm. Then $\bar{x}$ is a $\delta$-stationary point of the optimization problem (1).*

*Proof.* By Theorem 3.5, every $x_{k+1}$ found by running Equation (5) until convergence is the stationary point of the augmented Lagrangian, *i.e.* $\nabla\hat{\mathcal{L}}(x_{k+1}, z_k, u_k) = 0$. This gives us the standard update rule of classical ADMM, where the results of Huang et al. (2021) can be adapted directly. □

# B. Experimental Details

We present various details of our experimental setup. All experiments are run across three different seeds, and the results are provided as the mean and the standard error.

## B.1. Hyperparameters

Table 5: Hyperparameter details used/searched for SAFE and SAFE$^+$. Here, perturbation radius and dual interval were searched only in ResNet-20/CIFAR-10 and LLaMa-2-7b, then applied across all settings and target sparsity.

|  | **Vision** | **Language** |
|---|---|---|
| Epoch | 200 (ResNet20) or 300 (others) | 30 |
| Base optimizer | SGD | Adam |
| Batch size | 128 (or 126 when using 3 GPUs for data parallelism) | 8 |
| Learning rate | 0.1 | 0.0002 |
| Learning rate schedule | cosine | linear |
| Warm-up epoch | 5 | 2 |
| Weight decay | 0.0001 | 0 |
| Momentum | 0.9 | 0.9 |
| BNT sample size | 10000 | - |
| Perturbation radius ($\rho$) | $\{0.01, 0.05, \mathbf{\underline{0.1}}, 0.2, 0.5\}$ | $\{0.0001, \mathbf{\underline{0.0002}}, 0.0005, 0.01\}$ |
| Dual-update interval ($K$) | $\{1, 2, 4, 8, 16, \mathbf{\underline{32}}, 64, 128, 256, 512, 1024, 2048\}$ | $\{16, \mathbf{\underline{32}}, 64\}$ |
| Penalty parameter ($\lambda$) | $\{10^{-4}, 10^{-3}, 10^{-2}, 10^{-1}\}$ | $\{\mathbf{\underline{0.001}}, 0.005, 0.01, 0.05, 0.1\}$ |
| Penalty schedule | cosine warmup | constant |

Table 6: Best performing penalty parameter of SAFE for VGG-19 and ResNet-20/32.

| Sparsity | CIFAR-10 | | CIFAR-100 | |
|---|---|---|---|---|
| | VGG-19 | ResNet-20 | VGG-19 | ResNet-32 |
| 70~95% | $10^{-4}$ | $10^{-3}$ | $10^{-3}$ | $10^{-3}$ |
| 98% | $10^{-3}$ | | | |
| 99% | $10^{-2}$ | $10^{-2}$ | | $10^{-2}$ |
| 99.5% | | | | |

Across all experimental settings, the values for hyperparameters remain consistent, with the exception of the penalty parameter $\lambda$. We report these in Table 5. Basic hyperparameters such as learning rate, batch size, weight decay, and momentum are set to standard values commonly used in the literature (Kusupati et al., 2020; Ramanujan et al., 2020; Liu et al., 2019). For SAFE-specific hyperparameters, including perturbation radius and dual-update interval, values were optimized on ResNet-20/CIFAR-10, LLaMa-2-7b and applied universally across all settings, with only the penalty parameter $\lambda$ for image classification tasks being adjusted for each setting, which we report in Table 6. This demonstrates the general applicability of its hyperparameter values across different tasks.

Also, we use a cosine warmup schedule for the penalty parameter for the vision tasks, which increases the penalty parameter from 0 to the final target value with a cosine curve throughout training iterations.

### B.2. Experimental Details in Section 4.1

Here we train the standard 3-layer MLPs (with hidden layers of 300 and 100) on MNIST. For the sparsity experiment, we run standard dense training and SAFE with target sparsity of 90% and plot the distributions of weights. For the flatness measurements, we run SAFE and ADMM (Zhang et al., 2018) (a simple non-sharpness-aware baseline) with target sparsity of 90%. We then plot the loss landscape of the found solutions using standard visualization methods (Li et al., 2018), and compute their maximum Hessian eigenvalue (sharpness) using the power iteration method.

### B.3. Experimental Details in Section 4.2

We use standard ResNet and VGG architectures, with VGG having batch-norm layers instead of using dropout. For data augmentation, we used standard techniques such as random cropping and flipping. We used SGD for the base optimizer for SAFE and all baselines. All CIFAR experiments are conducted using a single or three NVIDIA RTX 3090, where a batch size of 126 was used in this case. For CIFAR-10/100, we trained ResNet-20 for 200 epochs, and ResNet-32 and VGG-19 for 300 epochs.

### B.4. Experimental Details in Section 4.3

Training data is processed following the standard settings in SparseGPT (Frantar & Alistarh, 2023), where we randomly sample 128 data points with a sequence length of 2048 from the first shard of C4 (Raffel et al., 2020). All experiments were conducted on a single GPU (NVIDIA A6000 or L40S) or HPU (Intel Gaudi2). We use LLaMa-2-7b-hf, LLaMA-2-13b-hf, and LLaMA-3.1-8 B models from the HuggingFace model hub (Wolf et al., 2020), implemented in PyTorch (Paszke et al., 2019). For SAFE and SAFE$^+$, we perform pruning over 30 epochs using the Adam (Kingma & Ba, 2017) optimizer as the base optimizer, with the hyperparameter $\beta$ set to (0.9, 0.95), without weight decay.

### B.5. Implementation and Reproduction Details

The code to reproduce the results of the paper is provided in `JAX`[2] (Bradbury et al., 2018; Heek et al., 2023) and `PyTorch`[3] (Paszke et al., 2019). Specifically, all image classification experiments using SAFE were conducted in `JAX`, while `PyTorch` was used for LLM experiments due to better support for official implementations and pretrained checkpoints of widely adopted models such as LLaMA. To obtain baseline performances, we either run our own implementations (ADMM, GMP, Magnitude) or official ones (SparseGPT, Wanda, ALPS), or refer to reported results from prior work (LTH from Wang et al. (2020); PBW and MLPrune from Zhou et al. (2021)). For sound comparison, when running our own or official implementations, we align key settings—such as model architecture and training epochs—with those used in prior works to produce reported performance.

---

[2] https://github.com/LOG-postech/safe-jax
[3] https://github.com/LOG-postech/safe-torch

## C. Detailed Results for Image Classification Tasks

We present precise numeric values for Figure 2 in Table 7.

Table 7: Validation accuracy of VGG-19 and ResNet-20/32 models pruned with SAFE and various baseline methods, trained on CIFAR-10 and CIFAR-100, across different sparsity levels. The results show that SAFE generally outperforms the baseline methods across all evaluated sparsity levels, indicating its robustness and effectiveness in maintaining accuracy even under high sparsity conditions.

| Dataset | Model | Method | Sparsity | | | | |
| --- | --- | --- | --- | --- | --- | --- | --- |
| | | | 90% | 95% | 98% | 99% | 99.5% |
| CIFAR-10 | VGG-19 | GMP | $93.37_{\pm0.09}$ | $93.13_{\pm0.12}$ | $93.08_{\pm0.09}$ | $92.70_{\pm0.19}$ | $90.63_{\pm0.14}$ |
| | | PBW | 93.87 | 93.57 | 92.83 | 90.89 | 10.00 |
| | | MLPrune | 93.70 | 93.45 | 92.48 | 91.44 | 88.18 |
| | | LTH | 93.51 | 92.92 | 92.34 | - | - |
| | | ADMM | $93.86_{\pm0.10}$ | $93.62_{\pm0.03}$ | $93.58_{\pm0.07}$ | $92.54_{\pm0.07}$ | $88.53_{\pm0.16}$ |
| | | SAFE | $\mathbf{94.65}_{\pm0.05}$ | $\mathbf{94.44}_{\pm0.08}$ | $\mathbf{94.05}_{\pm0.05}$ | $\mathbf{93.93}_{\pm0.17}$ | $\mathbf{93.56}_{\pm0.02}$ |
| | ResNet-20 | GMP | $92.94_{\pm0.08}$ | $91.81_{\pm0.14}$ | $89.42_{\pm0.17}$ | $85.15_{\pm0.19}$ | $75.83_{\pm0.47}$ |
| | | ADMM | $91.88_{\pm0.05}$ | $89.96_{\pm0.27}$ | $86.96_{\pm0.09}$ | $82.25_{\pm0.12}$ | $73.72_{\pm2.85}$ |
| | | SAFE | $\mathbf{93.44}_{\pm0.01}$ | $\mathbf{92.59}_{\pm0.09}$ | $\mathbf{89.58}_{\pm0.1}$ | $\mathbf{87.47}_{\pm0.07}$ | $\mathbf{79.55}_{\pm0.13}$ |
| CIFAR-100 | VGG-19 | GMP | $72.00_{\pm0.06}$ | $71.81_{\pm0.04}$ | $69.55_{\pm0.03}$ | $66.98_{\pm0.06}$ | $62.77_{\pm0.49}$ |
| | | PBW | 72.41 | 70.53 | 58.91 | 1.00 | 1.00 |
| | | MLPrune | 71.56 | 70.31 | 66.77 | 60.10 | 50.98 |
| | | LTH | 72.78 | 71.14 | 68.95 | - | - |
| | | ADMM | $72.93_{\pm0.07}$ | $71.17_{\pm0.16}$ | $70.02_{\pm0.34}$ | $67.23_{\pm0.34}$ | $43.40_{\pm0.71}$ |
| | | SAFE | $\mathbf{73.67}_{\pm0.21}$ | $\mathbf{72.83}_{\pm0.13}$ | $\mathbf{71.73}_{\pm0.09}$ | $\mathbf{70.02}_{\pm0.2}$ | $\mathbf{67.23}_{\pm0.19}$ |
| | ResNet-32 | GMP | $71.69_{\pm0.22}$ | $69.10_{\pm0.24}$ | $65.15_{\pm0.30}$ | $58.10_{\pm0.17}$ | $42.93_{\pm0.37}$ |
| | | PBW | 72.19 | 68.42 | 58.23 | 43.00 | 20.75 |
| | | MLPrune | 70.33 | 61.73 | 37.86 | 22.38 | 13.85 |
| | | LTH | 68.99 | 65.02 | 57.37 | - | - |
| | | ADMM | $70.85_{\pm0.45}$ | $68.74_{\pm0.31}$ | $63.75_{\pm0.06}$ | $49.13_{\pm0.22}$ | $12.34_{\pm0.73}$ |
| | | SAFE | $\mathbf{73.89}_{\pm0.24}$ | $\mathbf{72.33}_{\pm0.08}$ | $\mathbf{67.74}_{\pm0.24}$ | $\mathbf{62.77}_{\pm0.11}$ | $\mathbf{51.45}_{\pm0.32}$ |

## D. Additional Comparison with SAM-based pruners

Table 8: Comparison with other variants of CrAM on ResNet-20/CIFAR-10, where we also apply similar techniques to SAFE.

| Method | Sparsity | | | |
| --- | --- | --- | --- | --- |
| | 95% | 98% | 99% | 99.5% |
| CrAM | $90.18_{\pm1.80}$ | $69.53_{\pm12.36}$ | $45.17_{\pm20.86}$ | $10.00_{\pm0.00}$ |
| CrAM$^+$ | $93.62_{\pm0.06}$ | $91.75_{\pm0.41}$ | $88.82_{\pm0.18}$ | $81.30_{\pm0.56}$ |
| CrAM$_{\text{Multi}}$ | $92.21_{\pm0.79}$ | $91.05_{\pm0.34}$ | $88.83_{\pm0.19}$ | $84.82_{\pm0.27}$ |
| CrAM$^+_{\text{Multi}}$ | $92.17_{\pm0.91}$ | $91.06_{\pm0.32}$ | $88.98_{\pm0.04}$ | $84.95_{\pm0.38}$ |
| SAFE | $92.59_{\pm0.09}$ | $89.58_{\pm0.10}$ | $87.47_{\pm0.07}$ | $79.55_{\pm0.13}$ |
| SAFE$_{+\text{SG}}$ | $92.40_{\pm0.06}$ | $90.09_{\pm0.13}$ | $89.13_{\pm0.06}$ | $85.85_{\pm0.09}$ |
| SAFE$_{\text{Multi}}$ | $91.98_{\pm0.13}$ | $88.89_{\pm0.13}$ | $86.42_{\pm0.10}$ | $83.00_{\pm0.15}$ |
| SAFE$_{+\text{SG,Multi}}$ | $92.17_{\pm0.12}$ | $90.79_{\pm0.04}$ | $88.11_{\pm0.14}$ | $84.92_{\pm0.14}$ |

Table 9: Comparison with IMP+SAM on LLaMA2-7b for 50% sparsity.

| Method | Perplexity C4 / WikiText |
| --- | --- |
| IMP+SAM | 18.27 / 176.00 |
| SAFE | $\mathbf{8.91 / 6.79}$ |

### D.1. Other variants of CrAM

Here we compare SAFE with two additional variants of CrAM introduced in Peste et al. (2022): CrAM$_{\text{Multi}}$ and CrAM$^+_{\text{Multi}}$. Specifically, CrAM$_{\text{Multi}}$ additionally changes the target sparsity at each iteration chosen randomly from a predefined set, whereas CrAM$^+_{\text{Multi}}$ combines this with CrAM$^+$ explained in Section 4.5. Similarly to Section 4.5, for better comparison, we

apply these "+" and "Multi" strategy to SAFE by adding the gradient computed at compressed point to the sharpness-aware gradient of SAFE as $\nabla f(x + \rho \nabla f(x)/\|\nabla f(x)\|) + \underline{\nabla f(C(x))}$, and changing the target sparsity every $z$-updates and for every $\nabla f(C(x))$ in the $x$-update, which we denote as $\overline{\text{SAFE}_{+\text{SG}}}$, SAFE$_{\text{Multi}}$, and SAFE$_{+\text{SG,Multi}}$ for applying all both.

As shown in Table 8, SAFE achieve competitive performance until 98% sparsity and outperforms CrAM in exterme sparsities, despite improved performance of CrAM from CrAM$_{\text{Multi}}$ and CrAM$^+_{\text{Multi}}$. In particular, while SAFE achieves reliable performance without much additional techniques, CrAM, by itself, performs poorly in all sparsities, depending heavily on various auxiliary techniques to drastically improve performance. This is further supported by SAFE$_{+\text{SG}}$, where similar sort of benefits are observed. However, we find that the strategy of additionally employing gradients from various projected points also generally effective in SAFE. Although these findings are intriguing and merit further study, a comprehensive analysis lies beyond the scope of this work and is left to future research.

### D.2. IMP+SAM on Language model pruning

We extend the experiment in Section 4.5 to LLM pruning, where we prune LLama2-7b to 50% sparsity by applying IMP+SAM to block-wise reconstruction error objective similarly to SAFE. Here, We similarly perform pruning every 5 epochs with sparsity increasing linearly or cubically over the same number of epoch as SAFE for fair comparison. As shown in Table 9, SAFE outperforms IMP+SAM in LLM tasks similarly to results in image classification.

## E. Computation Cost Analysis LLM Pruning

We provide theoretical and empirical analysis of the computation costs of SAFE and various baselines in LLM pruning.

Table 10: Time complexity analysis for LLM post-pruning techniques

| Method | Time Complexity | Explanation |
|---|---|---|
| SparseGPT | $\mathcal{O}(L_B(Nd^2 + d^3))$ | Hessian computation over $N$ samples $(Nd^2)$ + Hessian inverse $(d^3)$ for $L_B$ layers in a single block |
| Wanda | $\mathcal{O}(L_B(Nd + d^2))$ | Activation norm computation $(Nd)$ + weight multiplication $(d^2)$ for $L_B$ layers in a single block |
| ALPS | $\mathcal{O}(L_B(N^2 + d^3 + kd^3))$ | Hessian computation over $N$ samples $(Nd^2)$ + eigendecomposition $(d^3)$ + penalized inverse for $k$ ADMM iterations $(kd^3)$ for $L_B$ layers in a single block |
| SAFE | $\mathcal{O}(L_B bkd^2)$ | Backpropagation through $L_B$ layers in a single block for $k$ iterations |

Table 11: Wall-clock time

| Method | Time (s) |
|---|---|
| Magnitude | 0.48 |
| Wanda | 3.98 |
| SparseGPT | 15.82 |
| ALPS | 788.66 |
| SAFE | 310.68 |

### E.1. Theoretical Time Complexity

We compare the time complexity for pruning a single transformer block with SAFE, SparseGPT, Wanda, and ALPS in terms of hidden dimensions $d$, number of data $N$ or batch size $b$, number of iterations $k$, and number of layers in a single block $L_B$ to observe how the computation cost scales. The results are given in Table 10.

We observe that while the computation cost of SAFE, similarly to ALPS, scales with the iteration, it does not scale with the size of the dataset $N$. Also, it only scales quadratically with the hidden dimension of the model. In comparison, SparseGPT and ALPS scale cubically. Given that scaling model and data sizes are a central strategy in the development of large language models, this highlights the advantage and adequecy of SAFE in the current era of large-scale models.

### E.2. Wall-clock Time

We report the wall clock time required for each pruning method on the LLaMA-2-7B model at 50% sparsity. Specifically, Table 11 shows the time taken to prune the first transformer block consisting of the self-attention and the feed-forward module. All measurements were conducted on a single Nvidia A6000 GPU to ensure a fair comparison across pruning methods.

# F. Ablation Study

We conduct ablation of various hyperparameters of SAFE on ResNet-20 trained on CIFAR-10.

### F.1. Effects of Penalty Parameter $\lambda$

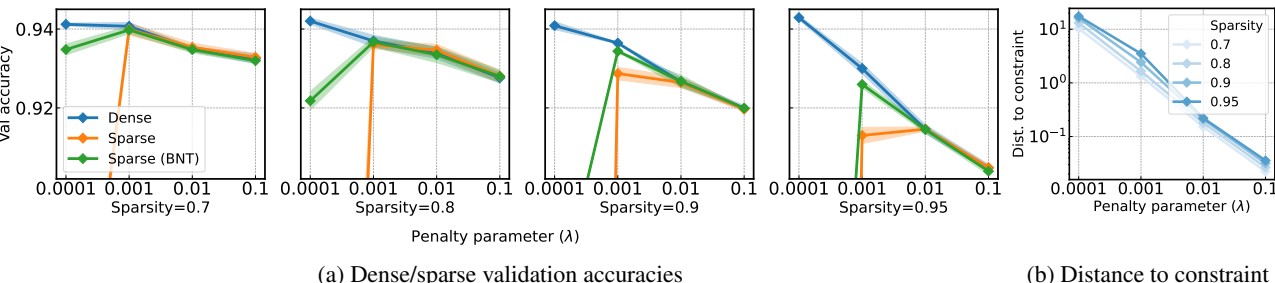

(a) Dense/sparse validation accuracies         (b) Distance to constraint

Figure 3: Effect of the penalty parameter $\lambda$ on final validation accuracy of dense/sparsified models (a) and the distance from the constraint (b) over various levels of sparsity. Larger $\lambda$ relieves the performance drop in the final projection step while degrading the performance of the original dense model. Also, BNT provides larger benefits for smaller $\lambda$ and the target sparsity

We observe how the penalty parameter $\lambda$ impacts various aspects of the final model in terms of the validation accuracy of the final network before and after projection (denoted as dense/sparse model in the legend, respectively) and the distance to the constraint. We vary $\lambda$ between $\{10^{-4}, 10^{-3}, 10^{-2}, 10^{-1}\}$ and observe this for target sparsity of $\{0.7, 0.8, 0.9, 0.95\}$, with is reported in Figure 3. We find that while larger $\lambda$ relieves the performance degradation in the final projection step by pushing the network closer to the sparsity constraint during training, it also degrades the performance of the original dense model. This indicates that a balance between objective minimization and constraint satisfaction would yield the best results. We also find that the model becomes more sensitive to $\lambda$ for higher target sparsity, where the degree to which large $\lambda$ incurs performance degradation while small $\lambda$ results in failure of the model to get sufficiently close to the constraint becomes much more severe. This highlights the challenge of training a model with extreme sparsity, which demands a careful balance between minimizing the objective function and satisfying the sparsity constraint.

### F.2. Effects of Batch-norm Tuning

We also observe how re-evaluating the batch statistics (*i.e.*, batch-norm tuning or BNT) for the final projected parameters impacts the performance of sparse networks over various sparsities. Our assumption is that it should be helpful when the parameters change greatly after projection, which will cause the hidden features to deviate from the statistics computed during training. The results are presented in Figure 3. We observe that the benefits of BNT are pronounced when $\lambda$ is small, where the distance to the constraint is the largest, fitting our assumption. However, we also observe that it fails to recover the performance in higher sparsity, despite having a similar distance to the constraint. We suspect it is due to the degraded quality of the sparse network having more impact on the performance in high sparsity, rather than the misalignment of the batch-norm statistics.

### F.3. Effect of $\lambda$ Scheduling

There are many strategies employed by the community known to boost performance for deep neural network training. One such strategy is scheduling, which has been a de facto for important hyperparameters such as learning rate (Goodfellow et al., 2016). In particular, gradual sparsity schedules (Zhu & Gupta, 2017; Benbaki et al., 2023) have been widely adopted to enforce less sparsity in the initial phases of training. Here we observe whether a similar effect can be transferred to the penalty parameter $\lambda$, an important parameter for controlling how strongly to push toward the sparsity constraint at any point during training. Precisely, we test whether a slow increase from zero to the targeted penalty $\lambda_f$ will yield improvements over constant $\lambda$. We train ResNet-20 on CIFAR-10 to $\{70\%, 80\%, 90\%, 95\%\}$ sparsity with SAFE using linear, cosine schedules, and constant $\lambda$, and observe how this affects the final accuracy of the sparse network in Table 12. We find that scheduling consistently yields overall higher accuracy, especially in extreme sparsity where it yields $+2\%$ increase.

To gain further insight as to how this occurs, we analyze how scheduling affects various aspects of training through the

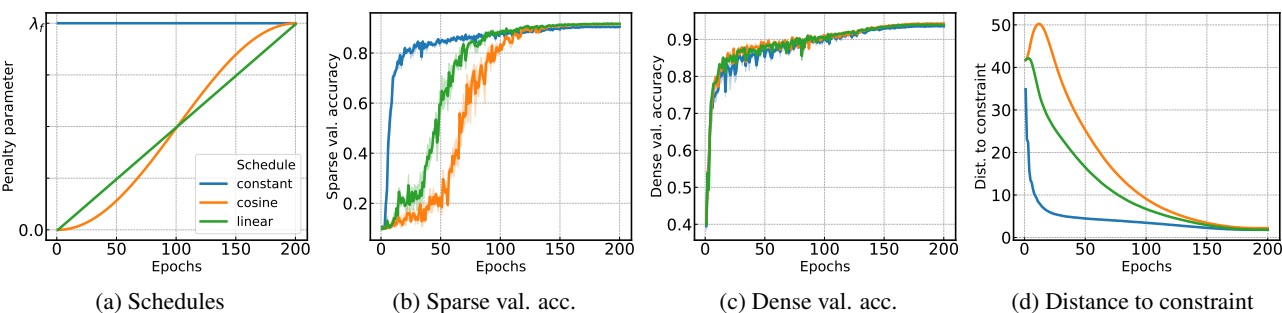

(a) Schedules     (b) Sparse val. acc.     (c) Dense val. acc.     (d) Distance to constraint

Figure 4: Effects of different choices of penalty parameter schedules (a) on validation accuracy of sparsified/dense network (b-c) and the distance to the target sparsity constraint (d) over the training process of ResNet-20/CIFAR-10 using SAFE on 95% sparsity. It is observed that scheduling yields better performance, seemingly allowing the network to move away from the constraint in the initial phases to focus more on training, which potentially underscores its impact on securing performance.

Table 12: Validation accuracy for various penalty parameter $\lambda$ schedules on ResNet-20 trained on CIFAR-10. The use of scheduling generally improves the final accuracy of the sparse network, especially at extreme sparsity.

| | Sparsity | | | |
| --- | --- | --- | --- | --- |
| | 70% | 80% | 90% | 95% |
| Constant | $93.79_{\pm 0.16}$ | $93.33_{\pm 0.10}$ | $92.13_{\pm 0.13}$ | $90.78_{\pm 0.16}$ |
| Linear | $93.78_{\pm 0.04}$ | $93.66_{\pm 0.17}$ | $93.06_{\pm 0.04}$ | $92.20_{\pm 0.01}$ |
| Cosine | $93.98_{\pm 0.09}$ | $93.67_{\pm 0.13}$ | $93.44_{\pm 0.01}$ | $92.59_{\pm 0.09}$ |

validation accuracy of the nearest sparse network (sparse val. acc.), the validation accuracy of the dense network (dense val. acc.), and the distance to the constraint in Figure 4. Here, we observe that while constant penalty pushes the network drastically close to sparsity in the initial stages, scheduling allows the network to temporarily stray away from the constraint. This seems to highlight that the initial phase of training is important for securing the performance of the final sparse network.

## F.4. Effects of dual-update interval $K$

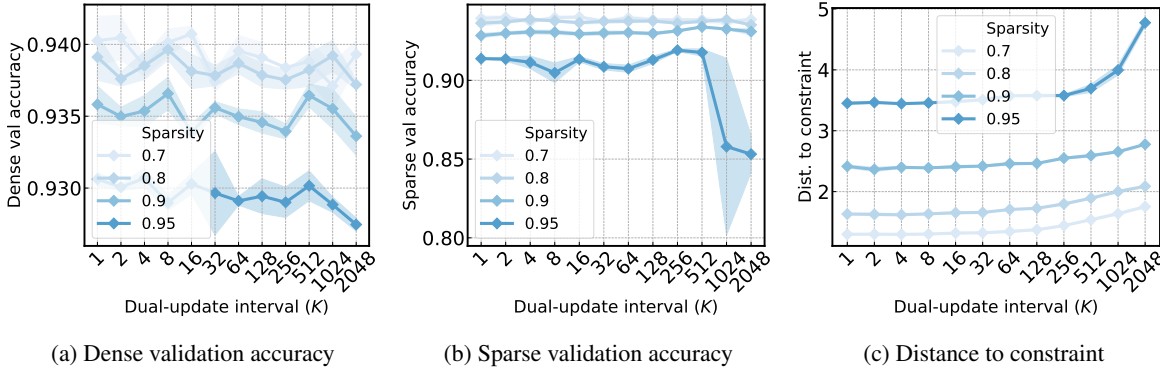

(a) Dense validation accuracy     (b) Sparse validation accuracy     (c) Distance to constraint

Figure 5: Effect of the dual-update interval $K$ on final validation accuracy of dense/sparsified models (a, b) and the final distance from the constraint (c) over various levels of sparsity. In our search range, $K$ has little impact on accuracy and distance to the constraint. However, in target sparsity of 95%, large $K$ fails to sufficiently push the network towards the sparsity constraint, resulting in performance degradation on the sparsified network.

We observe how the dual-update interval $K$ impacts the final validation accuracy before and after projection (denoted as

dense/sparse val accuracy, respectively) and the distance to the constraint in Figure 5. We find that within our search range, $K$ has little impact on the final network. However, in the case of target sparsity of $95\%$, we can observe that large $K$ fails to sufficiently push the network towards the sparsity constraint, resulting in performance degradation of the sparsified network. We can expect this trend to appear for all sparsity levels under increasing values of $K$, since this will result in less execution of dual ascent iteration for constraint satisfaction.

