# OpenReview forum: "SAFE: Finding Sparse and Flat Minima to Improve Pruning"
_ICML.cc/2025/Conference — ICML 2025 spotlightposter_

### Official Review · Reviewer_BN7C · 2025-02-25

**Overall Recommendation:** 4

**Summary:**

This work tried to learn sparse and flat minima during training, so such sparse and flat minima can be more suitable for network pruning. The idea is simple but effective. This work formulates pruning as a sparsity-constrained optimization problem where flatness is encouraged as an objective. This optimization problem can be solved explicitly via an augmented Lagrange dual approach, which naturally resulted in novel pruning methods called SAFE and its extension, SAFE+. Convergence analysis is presented. Empirical analysis include vision models and large language models. The improvements are significant.


## update after rebuttal

Thanks for the responses. It help addressing my concerns. I maintain the rating as Accept.

**Claims And Evidence:**

The claim and evidence are basically convincing.

**Essential References Not Discussed:**

An ICML2022 work Sparse Double Descent reveal some insights on generalization of network pruning, which may be helpful for understanding generalization of sparse minima.

[1] He, Z., Xie, Z., Zhu, Q., & Qin, Z. (2022, June). Sparse double descent: Where network pruning aggravates overfitting. In International Conference on Machine Learning (pp. 8635-8659). PMLR.

**Experimental Designs Or Analyses:**

I checked the experimental design and analyses. They make sense to me.

**Methods And Evaluation Criteria:**

Yes. The method and its motivation are clear and make sense. The evaluation criteria is common.

**Other Comments Or Suggestions:**

No.

**Other Strengths And Weaknesses:**

Strengths:

- SAFE that explicitly constrain sparsity and encourage flatness is novel to me.
- I appreciate the theoretical results for SAFE, while some minor concern exists.
- The empirical analysis is convincing.


Weaknesses:
- This work lacks efficiency analysis. Is SAFE computationally expensive compared with the baselines?
- In convergence analysis, Assumption 3.3 is a bit strong for deep loss landscape. Is it possible to prove the convergence for nonconvex optimization?

**Questions For Authors:**

Please see the weakness.

**Relation To Broader Scientific Literature:**

This paper make a key contribution to network pruning. Because, to my knowledge, it is the first successful method that can explicitly find sparse and flat minima in the training phase.

**Theoretical Claims:**

This work include a few theoretical results, particularly including deriving SAFE and its convergence analysis.

I can basically understand the proof sketch, but I did not carefully check the proof in the appendix.

In convergence analysis, Assumption 3.3 is a bit strong for deep loss landscape. Is it possible to prove the convergence for nonconvex optimization?

---

> ### Author Rebuttal · Authors · 2025-04-01
>
> We sincerely appreciate the reviewer’s recognition and constructive feedback. We have addressed the reviewer’s specific comments below, while we remain open to any additional suggestions.
>
>
> ---
>
> **Convergence for nonconvex case?**
>
> We appreciate the reviewer’s insight. We believe that the convergence of SAFE without the weak-convexity assumption can potentially be established by extending existing non-convex convergence analyses of sharpness-minimization [1, 2] and ADMM [3, 4] to SAFE’s x-minimization step and its sparsity-constrained alternating dual updates, respectively. We plan to explore this direction and aim to include the results in the revised version.
>
> ---
>
> **Lack of efficiency analysis**
>
> We appreciate the reviewer’s concern regarding the lack of efficiency analysis.
>
> First, we already provide wall-clock time comparisons between LLM pruning methods in Appendix C.1.
>
> In addition, we provide an in-depth time complexity analysis in our response to Reviewer `FtwJ` (“Wall clock time”) and discuss the operational overhead in response to Reviewer `h6df` (“Computational efficiency of SAFE”). While both SAFE and ALPS are iterative methods that introduce additional computation, SAFE is significantly more scalable than ALPS, and this cost is worthwhile trade-off given the favorable improvements in pruning quality.
>
>  Lastly, for vision tasks, all baseline methods except MLPrune(a one-shot method) rely on stochastic gradient-based optimization, and therefore share the same asymptotic time complexity as SAFE. While SAFE incurs additional cost compared to GMP and ADMM due to sharpness minimization, this overhead is offset by the consistently strong performance it has shown in our experiments across various tasks and sparsity levels.
> We will include this analysis into a dedicated cost analysis section in the revised version.
>
> ---
>
> **Essential references not discussed**
>
> Thank you for bringing this work to our attention. We present detailed discussions below.
>
> [5] presents an extensive empirical study on the generalization performance of various pruning approaches under label noise, revealing a double-desent phenomenon [6] across varying levels of sparsity. Intriguingly, in our label-noise experiments (Table 2), this double-descent behavior isn’t observed in SAFE, whereas it appears prominently in the ADMM baseline, which also consistently shows poor performance under label noise. We suspect this to be attributed to the strong regularization of sharpness minimization in mitigating label noise, as such effective regularization is known to mitigate double descent.
>
> Additionally, [5] makes a fascinating observation that higher sparsity tends to yield solutions of higher sharpness, which could potentially support our sharpness-minimization approach to improving pruning.
>
> We will make sure to include these discussions in the final version of the paper.
>
>
> ---
>
> **References** \
> [1] Khanh et al., Fundamental Convergence Analysis of Sharpness-Aware Minimization, NeurIPS 2024 \
> [2] Oikonomou and Loizou, Sharpness-Aware Minimization: General Analysis and Improved Rates, ICLR 2025 \
> [3] Ganzhao Yuan, Admm For Nonconvex Optimization Under Minimal Continuity Assumption, ICLR 2025 \
> [4] Huang and Chen, Mini-Batch Stochastic ADMMs for Nonconvex Nonsmooth Optimization, CoRR 2018 \
> [5] He et al., Sparse Double Descent: Where Network Pruning Aggravates Overfitting, ICML 2022 \
> [6] Nakkiran et al., Deep Double Descent: Where Bigger Models and More Data Hurt, ICLR 2020 \
> [7] Nakkiran et al., Optimal Regularization Can Mitigate Double Descent, ICLR 2021

---

### Official Review · Reviewer_gjwA · 2025-03-11

**Overall Recommendation:** 3

**Summary:**

This paper introduces SAFE and SAFE+ algorithms for finding sparse subnetworks with flat loss landscapes. The methods utilize an augmented Lagrange dual approach, with SAFE+ extending the base algorithm through a generalized projection operation. The authors evaluate their approaches on image classification and language modeling tasks, demonstrating that the resulting sparse networks achieve better generalization performance.

## Update after rebuttal
I appreciate the author's acknowledgment of my concerns in the future work section. However, since the current version does not directly address these points, I will maintain my score.

**Claims And Evidence:**

I have two concerns regarding the theoretical analysis in this paper:
1. While the authors claim that SAFE converges to the stationary point of the sparsity-constrained optimization problem under Assumptions 3.1-3.3, it is unclear why the convergence analysis appears to be independent of the sparsity ratio. This seems counterintuitive given the significant role of sparsity constraints in the optimization problem.
2. Despite the paper's emphasis on finding flat minima, which is explicitly incorporated into the learning objective, there appears to be no theoretical analysis or empirical evidence demonstrating the flatness of the obtained solutions.

**Essential References Not Discussed:**

The paper has appropriately cited and discussed the key related works.

**Experimental Designs Or Analyses:**

The experimental designs are sound.

**Methods And Evaluation Criteria:**

1. The core methodology employs an augmented Lagrangian dual approach to address sparsity-constrained optimization. Through alternating minimization, this formulation effectively decomposes complex optimization into tractable subproblems, demonstrating a sound implementation.
2. The SAFE+ variant incorporates an importance matrix P to enhance the sparsification process, enabling more informed pruning decisions by considering parameter significance.
3. What are the theoretical differences between SAFE and CrAM, and what mechanisms enable SAFE to achieve superior performance?

**Other Comments Or Suggestions:**

N/A

**Other Strengths And Weaknesses:**

N/A

**Questions For Authors:**

N/A

**Relation To Broader Scientific Literature:**

This work investigates neural network sparsification in conjunction with SAM, proposing an augmented Lagrangian approach that offers a theoretically grounded alternative.

**Theoretical Claims:**

While the proof demonstrates convergence to stationary points under specified assumptions, the relationship between convergence properties and different sparsity constraints remains unclear.

---

> ### Author Rebuttal · Authors · 2025-04-01
>
> We sincerely appreciate the reviewer for taking the time to review our work and providing us with constructive feedback. While we respond to the reviewer’s specific comments below, we would be keen to engage in any further discussion.
>
> ---
>
> **No sparsity ratio in our convergence analysis?**
>
> We appreciate this interesting question. In our algorithm, we operate by putting the iteration within the sparsity constraint through a tractable projection operation regardless of the sparsity ratio, so “whether convergence occurs at the limit of the algorithm” is guaranteed regardless of the sparsity ratio. However, the convergence *rate* may indeed be affected by the sparsity ratio, as certain factors related to it can influence the actual speed of convergence. This has been studied in the constrained optimization literature [1, 2, 3]. We believe exploring this aspect would be a valuable direction for future work to deepen our understanding of SAFE.
>
>
>
> ---
>
> **Theoretical analysis or empirical evidence of solution flatness**
>
> We would first like to clarify that we already provide empirical evidence that SAFE yields flatter solutions through Hessian eigenvalue measurements and loss landscape visualizations in Figure 1(c-d).
>
> However, we agree that a theoretical analysis of the flatness of the solution found by SAFE would be a valuable addition to the paper. It is well-established that the sharpness minimization used in SAFE yields flatter solutions [4-7]. Among these analyses, we plan to build on linear stability analysis [7] to derive an upper bound on the Hessian at the limit point of SAFE iterates, which would allow us to characterize the flatness of SAFE’s solution.
>
> We plan to explore in this direction and will aim to include it in the final version.
>
> ---
>
> **Theoretical difference between SAFE and CrAM**
>
> Thank you for the interesting question. While SAFE aims to find sparse and flat solution, CrAM aims to obtain a “compressible (dense) model” by enforcing a model to maintain a strong post-compression performance even with small perturbations:
>
> $$\min_x \max_{\|\epsilon\| \leq \rho}  f( C(x+\epsilon)  ).$$
>
> Based on this objective, CrAM updates the parameters as follows:
>
> $$x_{t+1} = x_t - \eta\nabla f( C(x+\rho\nabla f(x)) ),$$
>
> where the gradient is computed at a point obtained by applying a small perturbation followed by a hard compression step. However, this introduces an abrupt discontinuity between the iterate and the point where the gradient is evaluated, potentially disrupting the training process.
>
> On the contrary, SAFE progressively introduces sparsity into the optimization process via  augmented lagrangian and split variables in the ADMM framework, which may offer a more natural fit for neural network training.
>
> ---
>
>
> **References** \
> [1] Liu et al., One-bit compressive sensing with projected subgradient method under sparsity constraints. IEEE Transactions on Information Theory, 2019 \
> [2] Yuan et al., Dual Iterative Hard Thresholding, JMLR, 2020 \
> [3] Yuan et al., Smoothing Proximal Gradient Methods for Nonsmooth Sparsity Constrained Optimization: Optimality Conditions and Global Convergence, ICML, 2024 \
> [4] Wen et al., How Does Sharpness-Aware Minimization Minimize Sharpness?, ICLR 2023 \
> [5] Zhou et al., Sharpness-Aware Minimization Efficiently Selects Flatter Minima Late In Training \
> [6] Agarwala and Dauphin, SAM operates far from home: eigenvalue regularization as a dynamical phenomenon \
> [7] Shin et al., Critical Influence of Overparameterization on Sharpness-aware Minimization

---

### Official Review · Reviewer_h6df · 2025-03-11

**Overall Recommendation:** 4

**Summary:**

The authors tackle the problem of performance degradation in sparse neural networks at very high sparsity levels due to reduced capacity. The proposed method is motivated by the recent discoveries connecting the flatness of the solution (loss) landscape and the improved generalization performance of neural networks.

The authors frame the problem as an optimization problem that is aware of the sharpness of the loss landscape, and sparsity is used as a constraint. The set-up follows the standard SAM set-up [Foret2021], that is a min max problem for the loss function and a point in its perturbed surrounding. This causes the optimization process to explore a more flatter minima. An augmented Lagrangian approach is utilized that provides the smoothness of the Lagrangian for constrained optimization and allows for sparsity to be constrained as a projection function minimization. Finally, update equations are derived for each of the parameters.

The authors also introduce generalized projection that modifies the distance measure between the model weights and the projected weights. This allows for more sophisticated functions to be utilized as the pruning saliency metric. The results show that the proposed method SAFE (+) results in flatter minima as compared to ADMM. Further, SAFE results in better performance as compared to a wide range of baseline methods on tasks from two domains. Finally, the experiments also show that the proposed method is robust to label noise and one-shot sparsification to a higher sparsity ratio.

**Claims And Evidence:**

The claims made are very well supported by the clear and concrete results.

It would be also great to see the sharpness analysis using the hessian for all the other networks and tasks considered.

**Essential References Not Discussed:**

The paper discusses all the relevant ideas and methods from prior work in detail.

**Experimental Designs Or Analyses:**

The experimental designs and analysis are well thought out and show clear results.

**Methods And Evaluation Criteria:**

The proposed method, along with the motivation and derivations are sound. The evaluation is also extensive. However, the authors should consider comparison with additional methods that are closely related and have a similar objective of finding flatter minima to establish better performance over other methods with a similar objective. Finally, the results for [Peste2022] can be merged with Figure 2 for a better consolidation.

**Other Comments Or Suggestions:**

Minor: Figure 2b is missing other baselines.

**Other Strengths And Weaknesses:**

Strengths:

* The paper is very well written with clear motivation, method description and results.

Please see the sections above for additional strengths.

Weaknesses:

* One major concern is regarding the computational efficiency of the proposed method. Each iteration requires computation of multiple variables, that adds to the operations count. I would ask the authors to please discuss this in the final version of the paper.

Please see the sections above for other weaknesses.

**Questions For Authors:**

* The authors report the validation accuracies in the case of classification tasks. Are those accuracies computed at the last iteration ? or are those used during BNT?

## Update after rebuttal

I thank the authors for their response and for generating the requested results. Although I believe the paper makes several contributions, I will stick with my current rating as the rebuttal response does not make substantial improvements.

**Relation To Broader Scientific Literature:**

The paper builds upon the idea of achieving flatter minima for better generalization. This has been explored extensively in the case of dense networks and more recently for neural network sparsification as well. The major contribution of the work is in the set-up of the optimization process for both obtaining a flatter minima and constraining for sparsity.

**Theoretical Claims:**

The derivations made in the main body of the paper are sound. I was unable to spend a lot of time on the proofs presented in the Appendix section.

---

> ### Author Rebuttal · Authors · 2025-04-01
>
> We really appreciate the reviewer’s insightful comments and constructive feedback. While we address specific comments below, we would be keen to engage in any further discussions.
>
> ---
>
> **Extended sharpness measurements**
>
> Thank you for the insightful suggestion. We extend our sharpness analysis by computing the largest eigenvalue of the Hessian matrix in vision tasks. We use ResNet-20/32 on CIFAR-10/100 and compute the Hessian of the cross-entropy loss. The resulting sharpness is compared to that of the ADMM baseline.
>
> | Method | ResNet-20 / CIFAR-10 | ResNet-32 / CIFAR-100 |
> |-|-|-|
> | ADMM | 72.17 | 103.20 |
> | SAFE | **33.02**| **50.57** |
>
> Building upon the sharpness analysis in Section 4.1, we observe that SAFE continues to produce flatter solutions compared to ADMM when scaling up to deeper convolutional networks such as ResNet-20/32. These results support our claim that SAFE consistently finds sparse and flat solutions. We focused on vision architectures due to the rebuttal time limit and will expand the analysis in the revised version.
>
> ---
>
> **Comparison with other flat pruner**
>
> Thank you for the valuable suggestion. In response to the reviewer's comment regarding the need to compare with objectives that encourage flat minima, we implemented the method proposed by Na et al., [1], which integrates Sharpness-Aware-Minimization (SAM) into Iterative Magnitude Pruning (referred to as SAM+IMP). We applied SAM+IMP to both vision and LLM pruning tasks under our experimental setup, and compared its performance with that of SAFE.
>
>
> For both vision and LLM tasks, we apply SAFE and SAM+IMP using the same training budgets and hyperparameters as reported in the paper, to ensure a fair comparison. Pruning is performed periodically (every 10 epochs for vision / every 5 epochs for LLM) for SAM+IMP, with sparsity increasing linearly over time. We also consider a cubic sparsity schedule for SAM+IMP in the vision setting. In the LLM task, both methods employ the block-wise reconstruction error objective introduced in our work.
>
> **Vision**
> | Method | 95% | 98% | 99% | 99.5% |
> |-|-|-|-|-|
> | SAM+IMP (linear) | 80.30 ± 0.12  | 36.03 ± 4.19  | 18.30 ± 2.80  | 13.80 ± 0.52  |
> | SAM+IMP (cubic) | 92.50 ± 0.05  | 89.24 ± 0.06  | 83.74 ± 0.14  | 73.73 ± 0.30  |
> | SAFE | **92.59 ± 0.07**  | **89.58 ± 0.10**  | **87.47 ± 0.07**  | **79.55 ± 0.13**  |
>
> **LLM**
> | Method | PPL (C4) | PPL (WikiText2) |
> |-|-|-|
> | SAM+IMP | 18.27 | 176.00 |
> | SAFE | **8.91** | **6.79** |
>
> As shown in the results above, SAFE consistently outperforms SAM+IMP across vision and LLM tasks. We conjecture that the pruning operation in SAM+IMP is hard and abrupt, which may hinder effective sharpness minimization. In contrast, SAFE introduces a smooth penalization scheme via the augmented lagrangian, enabling joint optimization of sharpness and sparsity in a more balanced manner—leading to superior performance.
>
> ---
>
> **Computational efficiency of SAFE**
>
> Thank you for raising this concern. As the reviewer mentioned, ADMM-based approaches such as SAFE and ALPS introduce auxiliary variables ($z,u$), which adds to the overall operation count. However, unlike ALPS, SAFE updates these variables only once every $k$ iterations (we use $k=32$ in the paper), resulting in minimal overhead–less than a 3% increase in total operations.
>
> Moreover, the update cost of $z$ and $u$ is negligible compared to the weight update $w$. Specifically, $z$ requires $O(dlogd)$ for sorting, $u$ involves only element-wise addition $O(d)$, whereas updating $w$ via backpropagation incurs $O(d^2)$ complexity.
>
> We have also provided an in-depth time complexity comparison of various pruning algorithms in our response to Reviewer `FtwJ` . While SAFE has higher time complexity than one-shot methods due to its iterative nature, it generally achieves better pruning performance. This reflects a meaningful cost-performance tradeoff. We will include this efficiency analysis in the revised version.
>
> ---
>
> **Others**
>
> > “The authors report the validation accuracies in the case of classification tasks. Are those accuracies computed at the last iteration ? or are those used during BNT?”
>
> All the results in the paper for classification task report validation accuracy after final projection followed by batch-normalization training.
>
> > “[Peste2022] can be merged with Figure 2 for a better consolidation.”
>
> Thank you for the suggestion. We will incorporate this suggestion in the revised version.
>
> ---
>
> **References** \
> [1] Na et al., Train Flat, Then Compress: Sharpness-Aware Minimization Learns More Compressible Models, EMNLP 2022 findings

---

### Official Review · Reviewer_FtwJ · 2025-03-14

**Overall Recommendation:** 4

**Summary:**

This paper presents an algorithm to produce networks that are both flat and sparse. Recent work has demonstrated that flat networks have better generalization and have proposed some approaches for producing flat and sparse networks. This paper introduces a theoretically principled approach for producing sparse and flat models. The approach is based on an augmented Lagrange dual approach and is proven to converge. The proposed algorithm outperforms other pruning algorithms across several tasks, networks, and pruning strategies.

## Update after Rebuttal

The authors have answered my questions and I maintain a score of 4.

**Claims And Evidence:**

Yes. The authors demonstrate that their proposed algorithm does indeed lead to flatter and sparser solutions than standard training.

**Essential References Not Discussed:**

N/A

**Experimental Designs Or Analyses:**

The experimental design seems valid to me and I did not see any issues.

**Methods And Evaluation Criteria:**

Yes, the authors compare their method to relevant baselines and recent methods. They compare across a variety of tasks and architectures to comprehensively demonstrate the strong performance of their method. I would be interested in seeing the performance of their method under different types of noisy data (ie image corruptions such as ImageNet-C or adversarial robustness) in addition to label noise but I do not think this is necessary to make their point clear, especially if these experiments cannot be run during the rebuttal period.

**Other Comments Or Suggestions:**

- line ~370 in the right column has a missing table reference (‘see ??’)

**Other Strengths And Weaknesses:**

This paper explains the method very well, including relevant information in the main paper with necessary supplementary details in the appendix. I appreciate the clarity and exposition of the method and the comprehensiveness of the results. There are no critical weaknesses; I would like to see additional robustness experiments but understand the time constraint, and I would appreciate a further discussion of the complexity of the method but neither of these are necessary for completeness.

**Questions For Authors:**

Here I will restate the two main questions I would like to see addressed

1. Does this method additionally improve robustness to corruptions (ie ImageNet-C) or adversarial examples?
2. Could you provide further details about what leads to the high runtime of your algorithm and whether there are any ways to reduce this (similar to some of the works building on top of SAM to reduce its complexity via only performing the update every n steps)? SAM pointers: Sharpness-Aware Training for Free by Du et al; Towards Efficient and Scalable Sharpness-Aware Minimization by Liu et al; An Adaptive Policy to Employ Sharpness-Aware Minimization by Jiang et al.

**Relation To Broader Scientific Literature:**

This work builds upon a few previous papers in the space that motivate a need for flat and sparse neural networks. This paper proposes a theoretically principled method that also performs well on several tasks and outperforms existing methods.

**Theoretical Claims:**

The theoretical analysis makes sense to me and I did not see any issues.

---

> ### Author Rebuttal · Authors · 2025-04-01
>
> We sincerely thank the reviewer for finding our work interesting and giving us constructive suggestions. We make clarifications to specific comments below.
>
> ---
>
> **Robustness to different types of noisy data**
>
> Thank you for the insightful suggestion. To test the robustness of SAFE against image corruptions and adversarial attacks, we evaluate ResNet-20 models pruned to various sparsities with SAFE and ADMM on CIFAR-10C (due to limited time frame) and adversarial samples created using $l_\infty-PGD$ attack following [2], respectively.
>
> `CIFAR-10C (intensity=3)`
> | method | 90% | 95% | 98% | 99% | 99.5% |
> |:-|-:|-:|-:|-:|-:|
> | ADMM  | 70.06±0.03 | 68.92±0.26 | 65.45±0.29 | 59.19±0.47 | 55.71±0.45 |
> | SAFE  | **73.98±0.09** | **72.92±0.41** | **68.20±0.47** | **66.02±0.56** | **56.58±0.36** |
>
>
> `Adversarial noise (l∞-PGD)`
> | method| 90% | 95% | 98% | 99% | 99.5% |
> |:-|-:|-:|-:|-:|-:|
> | ADMM | 49.81±1.02  | 49.84±1.78  | 43.33±1.59  | 30.29±0.64  | 23.25±1.92  |
> | SAFE | **56.43±1.03** | **51.40±0.89** | **43.34±0.90** | **43.80±1.27** | **29.48±0.68** |
>
> Similar to label-noise experiments, SAFE consistently outperforms the baseline ADMM under these corruptions, indicating strong robustness of sharpness minimization toward various types of noise as suggested in [1,2]. We will include these results in the revised version.
>
> ---
>
> **Wall clock time**
>
> > Could you provide further details about what leads to the high runtime of your algorithm
>
> Thank you for your question. We analyze the main source of SAFE’s runtime overhead from both theoretical and engineering perspectives.
>
> From a theoretical perspective, we present the time complexity for pruning a single transformer block : $d$ = hidden dim, $N/b$ = # of data / batch size, $k$ = # of iterations,  $L$ = # of layers per block.
>
> | Method | Time Complexity | Explanation |
> |-|-|-|
> | SparseGPT | $O(L(Nd^2 + d^3))$ | Hessian over $N$ samples ($Nd^2$) + inverse ($d^3$) |
> | Wanda | $O(L(Nd + d^2))$ | Activation norm ($Nd$) + weight multiplication($d^2$) |
> | ALPS | $O(L(Nd^2 + d^3 + kd^3))$ | Hessian over $N$ samples ($Nd^2$) + eigendecomposition ($d^3$) + penalized inverse for $k$ iterations ($kd^3$) |
> | SAFE | $O(Lbkd^2)$ | Backprop through $L$ layers for $k$ iterations |
>
> Unlike one-shot methods (e.g., SparseGPT, Wanda), iterative methods (SAFE, ALPS) incur runtime overhead as iterations($k$) increase. However, SparseGPT scales cubically due to Hessian inversion, while SAFE’s quadratic complexity offers better efficiency. ALPS also suffers from poor scalability due to second-order computation, making SAFE more practical for large models.
>
> From an engineering perspective, CPU offloading was introduced for memory efficiency, but it also contributed to runtime overhead.
>
> > …any ways to reduce this (runtime)?
>
> To improve SAFE’s runtime, we can consider reducing: (1) iterations $k$, and (2) per-iteration cost.
>
> - **Fewer iterations**: Prior work has proposed ways to accelerate ADMM, such as adaptive penalties [3,4], over-relaxation [4,5], preconditioning [6], and variance reduction [7].
>
> - **Cheaper iteration**: Following the reviewer’s suggestion, SAFE can adopt efficient SAM variants [8,9] and further engineering optimizations.
>
> We applied one such optimization by moving data to GPU memory, reducing per-iteration cost. (see Appendix B.3 and C.1 for configuration details)
>
> | Config | Runtime (s) |
> |-|-|
> | CPU offloading | 460 |
> | GPU | 277 |
>
> Removing CPU offloading yielded a 40% speedup, with an additional ~8Gb GPU memory usage under SAFE’s configuration. This reflects a speed-memory tradeoff. Note that runtime differences may result from varying hardware.
>
> While we have not yet incorporated algorithmic acceleration techniques (e.g., ADMM improvements or SAM variants), we believe they hold promise for further reducing runtime, and we plan to explore them in future work.
>
> ---
>
> **typo (Line 370)**
>
> Thank you for pointing it out. We will have them fixed in the revised paper.
>
> ---
>
> **References** \
> [1] Wei et al., Sharpness-Aware Minimization Alone can Improve Adversarial Robustness., ICML 2023 workshop. \
> [2] Zhang et al., On the Duality Between Sharpness-Aware Minimization and Adversarial Training., ICML 2024 \
> [3]  Yuan et al., Admm For Nonconvex Optimization Under Minimal Continuity Assumption, ICLR 2025 \
> [4] Xu et al., Adaptive Relaxed ADMM: Convergence Theory and Practical Implementation, CVPR 2017 \
> [5] Song et al., Optimizing ADMM and Over-Relaxed ADMM Parameters for Linear Quadratic Problems, AAAI 2024 \
> [6] Ali et al., A Semismooth Newton Method for Fast, Generic Convex Programming, ICML 2017 \
> [7] Liu et al., Accelerated Variance Reduction Stochastic ADMM for Large-Scale Machine Learning., IEEE TPAMI 2020  \
> [8] Du et al., Sharpness-Aware Training for Free, NeurIPS 2022 \
> [9] Liu et al., Towards Efficient and Scalable Sharpness-Aware Minimization, CVPR 2022

---

### Decision · Program_Chairs · 2025-05-01

**Decision:**

Accept (spotlight poster)

**Comment:**

This paper introduces SAFE and its extension SAFE+, algorithms designed to jointly promote sparsity and flatness in neural networks. The methods are motivated by the strong empirical link between flatter minima and better generalization. Built upon an augmented Lagrangian dual formulation, the approach explicitly enforces sparsity constraints while incorporating sharpness-aware objectives, leading to the construction of sparse subnetworks that generalize well. Extensive empirical evaluations and theoretical grounding are provided.
The reviewers largely agree that this is a strong paper with theoretical depth, practical impact, and sound experimentation. While there are some open questions regarding efficiency and deeper flatness theory, the core idea is novel and well executed. The rebuttal was comprehensive and addressed reviewer concerns effectively, leading all reviewers to maintain or confirm high scores. Therefore, the AC suggests acceptance and encourages the authors to include the rebuttal discussion in the final version of the paper.